# Stochastic asymmetric repartition of lytic machinery in dividing CD8[+] T cells generates heterogeneous killing behavior

Fanny Lafouresse[1†]*, Romain Jugele[1†], Sabina Müller[1], Marine Doineau[2], Valérie Duplan-Eche[3], Eric Espinosa[1], Marie-Pierre Puisségur[1], Sébastien Gadat[2], Salvatore Valitutti[1,4]

[1]INSERM U1037, Centre de Recherche en Cancérologie de Toulouse (CRCT), Université de Toulouse III-Paul Sabatier, Toulouse, France; [2]Toulouse School of Economics, CNRS UMR 5314, Université Toulouse 1 Capitole, France and Institut Universitaire de France, Toulouse, France; [3]INSERM, UMR1043, Centre de Physiopathologie de Toulouse Purpan, Toulouse, France; [4]Department of Pathology, Institut Universitaire du Cancer-Oncopole de Toulouse, Toulouse, France

**Abstract** Cytotoxic immune cells are endowed with a high degree of heterogeneity in their lytic function, but how this heterogeneity is generated is still an open question. We therefore investigated if human CD8[+] T cells could segregate their lytic components during telophase, using imaging flow cytometry, confocal microscopy, and live-cell imaging. We show that CD107a[+]-intracellular vesicles, perforin, and granzyme B unevenly segregate in a constant fraction of telophasic cells during each division round. Mathematical modeling posits that unequal lytic molecule inheritance by daughter cells results from the random distribution of lytic granules on the two sides of the cleavage furrow. Finally, we establish that the level of lytic compartment in individual cytotoxic T lymphocyte (CTL) dictates CTL killing capacity.

*For correspondence:
fanny.lafouresse@inserm.fr

†These authors contributed equally to this work

Competing interests: The authors declare that no competing interests exist.

## Introduction

Heterogeneity and plasticity of lymphocyte function are key components of successful adaptive immune responses. Accordingly, several studies put forth the notion that individual mouse and human lymphocytes exhibit high degrees of heterogeneity in both their phenotypic and functional characteristics (*Beuneu et al., 2010*; *Buchholz et al., 2016*; *Buchholz et al., 2013*; *Ganesan et al., 2017*; *Kumar et al., 2018*; *Lemaître et al., 2013*; *Newell et al., 2012*). Functional heterogeneity is not limited to cell differentiation and acquisition of phenotypic and functional characteristics, but also involves late steps of immune cell responses such as CD8[+] cytotoxic T lymphocyte (CTL)- and natural killer cell-mediated cytotoxicity (*Guldevall et al., 2016*; *Halle et al., 2016*). Accordingly, we have previously shown that human CTL belonging to the same clonal population exhibit heterogeneity in their lytic function during sustained interaction with target cells (*Vasconcelos et al., 2015*). While some CTL kill a limited number of target cells, others emerge as super-killer cells.

One proposed mechanism of functional heterogeneity generation in T lymphocytes is asymmetric cell division (ACD). ACD is a key mechanism to generate cell heterogeneity in biology. It plays a crucial role in embryogenesis by allowing the formation of two distinct cells from a single mother cell (*Dewey et al., 2015*; *Knoblich, 2008*). In immunology, ACD has been proposed as a process allowing mouse naive T lymphocytes to divide into short-lived effector T cells and memory T cells, after TCR-triggered division (*Arsenio et al., 2015*; *Arsenio et al., 2014*; *Chang et al., 2011*; *Chang et al., 2007*).

In the present work, we investigated the possibility that, in dividing human CD8$^+$ T cells, heterogeneous distribution of molecules relevant for cytotoxic function into nascent daughter cells might contribute to CTL killing heterogeneity.

To address this question, we employed imaging flow cytometry, 3D confocal laser scanning microscopy, live-cell imaging, and mathematical modeling to investigate whether and how lytic components might differently segregate in telophase.

Our results show that both freshly isolated human peripheral blood CD8$^+$ T cells and clonal CTL exhibit a heterogeneous repartition of lytic machinery in telophase during TCR-triggered proliferation, which is not part of a classical ACD process. Furthermore, we demonstrate that heterogeneous lytic compartment repartition resets at each round of CTL division and is consequently stationary, but not hereditary. Finally, we show that the level of lytic granule expression in individual CTL influences their killing ability.

Together, our results unveil a mechanism of stochastic uneven repartition of pre-packaged lytic components within intracellular vesicles that generates functional plasticity during division and contributes to lytic function heterogeneity of individual cells belonging to clonal populations.

## Results

### Imaging flow cytometry reveals uneven repartition of lytic machinery in dividing human CD8$^+$ T cells

To investigate the mechanisms leading to the generation of CTL exhibiting heterogeneous killing ability, we first measured the distribution of lytic machinery components in dividing human CD8$^+$ T cells. Telophase is the bona fide cell cycle phase where unambiguous measurement of molecular repartition in nascent daughter cells is performed (*Chang et al., 2007*; *Filby et al., 2011*). Lytic granule repartition during human CD8$^+$ T cell division was evaluated using imaging flow cytometry, a technique that combines the advantages of both flow cytometry and microscopy (*Basiji and O'Gorman, 2015*; *Doan et al., 2018*; *Hritzo et al., 2018*). This approach allowed us to collect and analyze a substantial number of cells and to visualize and assess the repartition of molecules of interest within individual cells that were unambiguously identified as being in telophase. Cells in telophase were identified using a computer-assisted gating strategy, on the basis of nuclear and tubulin stainings (*Figure 1—figure supplement 1*). Nuclear staining with SYTOXorange identified bi-nucleated cells with elongated shape corresponding to cells in the late steps of division (anaphase and telophase). The cells in telophase were identified (and discriminated from possible cellular doublets) on the basis of tubulin staining that allowed us to highlight their midbodies. *Figure 1—figure supplement 2A* shows how masks were applied to delimit the cells and measure the fluorescence intensity of markers of interest in the nascent daughter cells. Cells were also stained with Cell Trace Violet (CTV), a probe that labels total cell proteins. As previously reported (*Filby et al., 2011*), we observed that total proteins distribute in nascent daughter cells within a range of 40–60% (*Figure 1—figure supplement 2B*). In our study, CTV staining served both as a marker of cell division (allowing us to identify cells in the different division rounds [*Quah and Parish, 2012*]), and to define total protein repartition in telophase (*Filby et al., 2011*). This procedure minimized the possibility that, if some images were taken slightly on an angle, with one daughter cell slightly more in focus than the other, the markers of interest would artificially appear as asymmetric. Indeed, asymmetric distribution was defined as cells in telophase in which repartition of the marker of interest in the nascent daughter cells was beyond the 40–60% limits observed for CTV repartition (*Figure 1—figure supplement 2B*). In addition, to further exclude the possibility of measurement artifacts, we verified individual cells by eyes and included in the analysis only cells in telophase that were on an even plane. Specificity of staining for the various markers was validated (see Material and methods).

In a first approach, CD8$^+$ T cells freshly isolated from healthy donor blood samples were stimulated with immobilized anti-CD3/anti-CD28/ICAM-1 for 72 hr. Anti-CD3/anti-CD28/ICAM-1 stimulation resulted in activation of human CD8$^+$ T cells as shown by cell proliferation and CD137 upregulation (*Figure 1—figure supplement 3*). Repartition of the lysosomal marker CD107a was investigated in cells in telophase. As shown in *Figure 1A*, while CTV distribution ranged between 40 and 60% in dividing T cells, 23% of telophasic CD8$^+$ T cells exhibited an uneven distribution of CD107a$^+$ vesicles overcoming the 40–60% CTV range.

We next investigated the distribution in telophase of lytic components such as perforin and granzyme B (GrzB), molecules known to be pre-stored in lytic granules. As shown in *Figure 1B,C*, perforin and GrzB also unevenly segregated into the two nascent daughter cells in telophase, indicating that daughter cells received a heterogeneous quantity of lytic components.

The slope of the linear regression curve for the distribution of CD107a, perforin, and GrzB as compared to CTV was close to 0.1, indicating that these three molecules distributed independently from total proteins.

To define whether uneven repartition of lytic components could be observed in fully differentiated cells, such as memory cells, we investigated CD107a and perforin distribution in telophase in purified human memory CD8$^+$ T cells. This analysis showed that also memory CD8$^+$ T cells exhibited uneven repartition of CD107a and perforin in telophase (*Figure 1—figure supplement 4*).

We next investigated whether lytic machinery asymmetric repartition could also be observed in activated CD8$^+$ T cell populations composed of monoclonal cells such as antigen-specific CTL clones. To address this question, we investigated CD107a repartition in CTL undergoing cell division. For this study, we activated CTL clones using immobilized anti-CD3/anti-CD28/ICAM-1 for 72 hr. We opted for this stimulation condition since, in preparatory experiments, we observed that conjugation of CTL with cognate target cells results (during the 72 hr culture) in the creation of cellular clumps and debris due to CTL killing activity, thus making it difficult and potentially misleading to analyze cells by image flow cytometry and conventional microscopy. As shown in *Figure 1D*, we observed that in clonal CTL undergoing cell division, 15% of the two nascent daughter cells in telophase exhibited uneven distribution of CD107a, thus confirming and extending observations obtained using CD8$^+$ peripheral blood T cells.

Taken together, the above results indicate that a lysosomal-associated membrane protein known to be a marker of lytic granules and effector molecules involved in CTL lytic function, unevenly segregate in 10–23% of individual human CD8$^+$ T cells undergoing division.

## Confocal laser scanning microscopy confirms uneven repartition of lytic machinery in dividing CD8$^+$ T cells

Image flow cytometry allows the unambiguously identification and capture of rare events within a cell population, such as cells in telophase, albeit exhibiting a lower resolution when compared to classical imaging methods. This notion prompted us to confirm results obtained using imaging flow cytometry, with additional methods.

We therefore used 3D confocal laser scanning microscopy to measure CD107a content in telophasic CD8$^+$ T cells following stimulation with immobilized anti-CD3/anti-CD28/ICAM-1. Although this approach allowed us to collect a relatively small number of cells in telophase (n = 61 compared to n = 908 obtained by image flow cytometry), it revealed that 27% of the CD8$^+$ T cells in telophase exhibited uneven repartition of CD107a, above a 1.5 threshold (corresponding to the 40–60% range used in imaging flow cytometry experiments) (*Figure 2A*). *Figure 2B* depicts the maximum intensity projection of a z-stack of images on which measurements of fluorescence intensity were performed (left panel) and a central z-section (right panel). The asymmetry of CD107a repartition in nascent daughter cells is better appreciated by looking at the 3D reconstructions of the dividing cell (*Figure 2—video 1*).

Together, the above results indicate that confocal laser scanning microscopy provides results that reinforce those we obtained using imaging flow cytometry and supports the finding that lytic granules undergo uneven repartition in ~20% of dividing CD8$^+$ T cells.

## Uneven repartition of lytic machinery is not accompanied by asymmetric segregation of fate determining transcription factors and does not require a polarity cue

The observation that lytic components were unevenly inherited in daughter cells prompted us to investigate whether this process was somehow related to mechanisms of cell fate determining ACD, a process reported to play a role in mouse naive T lymphocytes differentiation (*Arsenio et al., 2015*; *Arsenio et al., 2014*; *Kamiński et al., 2016*; *Pham et al., 2014*). Indeed, it has been reported that ACD can result in the generation of one daughter cell predisposed to become a short-lived effector cell (harboring a high level of the transcription factors T-bet and c-Myc, and of GrzB) and one

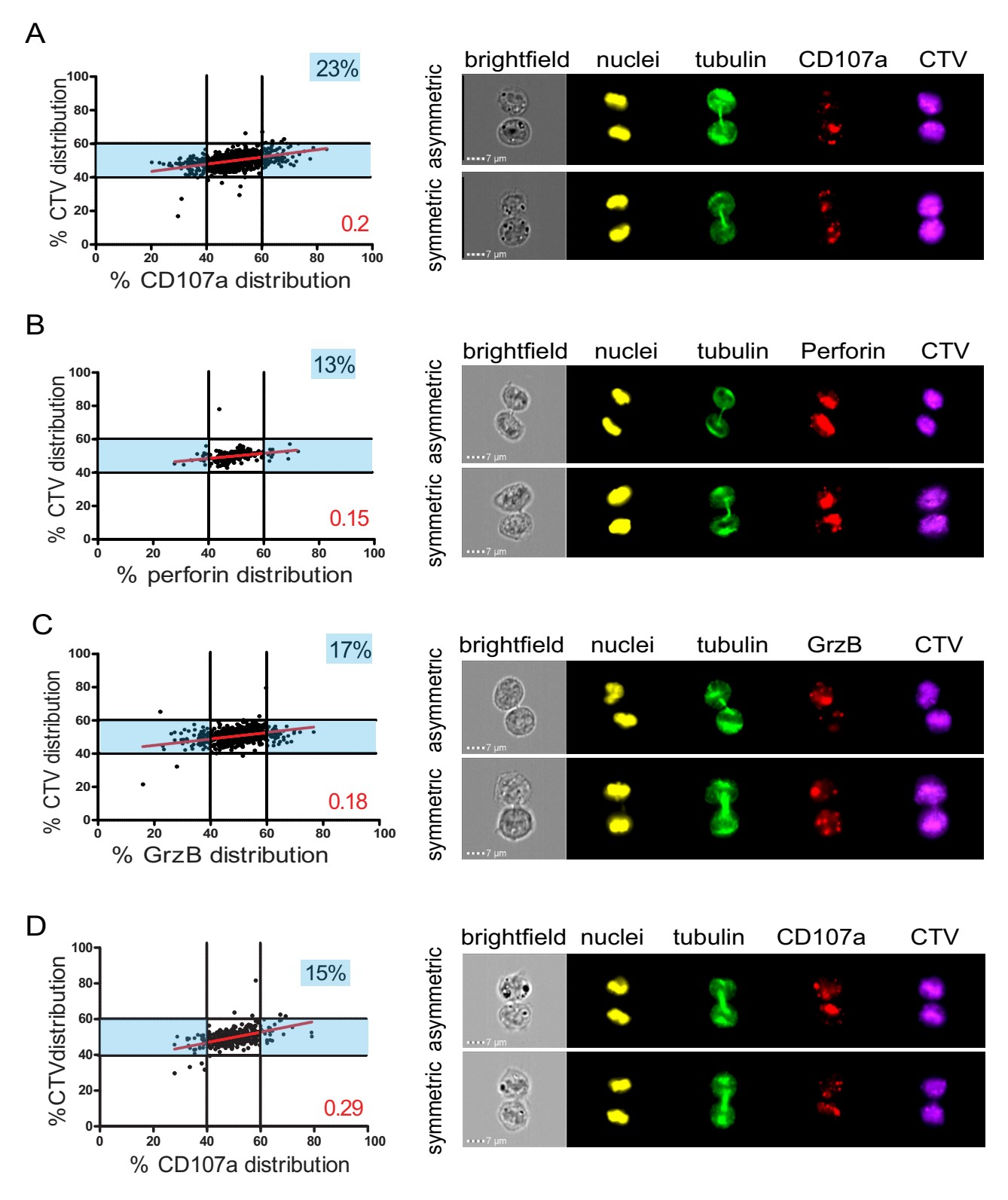

**Figure 1.** Lytic components are asymmetrically distributed in dividing CD8[+] T cells. (A–C) Freshly isolated polyclonal CD8[+] T cells or (D) CTL clones were stimulated by immobilized anti-CD8/anti-CD28/ICAM-1 during 72 hr and stained with antibodies directed against the indicated markers. Cells in telophase were identified using imaging flow cytometry. (A) Left panel: Each dot represents one nascent daughter cell. Only one of the two nascent daughter cells in telophase is plotted. The percentage of staining for CD107a in the presented cell (x axis) is plotted against the percentage of staining

*Figure 1 continued on next page*

*Figure 1 continued*

for total cell proteins (CTV, y axis). Asymmetric cells were defined as cells in telophase in which repartition of CD107a in the nascent daughter cells was beyond the 40–60% observed for CTV repartition (n = 908 from three independent experiments). Right panel: Example of asymmetric and symmetric cell distribution of CD107a, as detected by imaging flow cytometry. (B) Left panel: The percentage of staining for perforin in the presented nascent daughter cell is plotted as in (A). Asymmetric cells were defined as indicated in (A) (n = 191 from three independent experiments). Right panel: Example of asymmetric and symmetric cell distribution of perforin. (C) Left panel: The percentage of staining for GrzB in the presented nascent daughter cell is plotted as in (A). Asymmetric cells were defined as indicated in (A) (n = 728 from two independent experiments). Right panel: Example of asymmetric and symmetric cell distribution of GrzB. (D) Left panel: The percentage of staining for CD107a is plotted as in (A). Asymmetric cells were defined as indicated in (A) (n = 352 from three independent experiments). Right panel: Example of asymmetric and symmetric cell distribution of CD107a. Numbers highlighted in blue in the plots indicate the percentage of cells exhibiting asymmetric repartition of the marker of interest. Red lines indicate the global distribution of the data. Red numbers indicate the slope of the linear regression curve for marker distribution. See Figures S1, S2, S3, and S4.

The online version of this article includes the following source data and figure supplement(s) for figure 1:

**Source data 1.** CD107a, perforin, and granzyme B distribution between daughter cells.
**Figure supplement 1.** Gating strategy for imaging flow cytometry (IsX) acquisition.
**Figure supplement 2.** Analysis and representation of the repartition of markers of interest in dividing cells.
**Figure supplement 3.** CD8[+] T cells are efficiently stimulated on coated anti-CD3/anti-CD28/ICAM1.
**Figure supplement 4.** Uneven lytic granule segregation in telophase in CD8[+] memory T cells.

daughter cell predisposed to become a memory T cell (*Widjaja et al., 2017*). We investigated whether uneven repartition of fate determining transcription factors T-bet and c-Myc (*Chang et al., 2011*; *Verbist et al., 2016*) might occur in telophase in freshly isolated peripheral blood CD8[+] T cells stimulated with anti-CD3/anti-CD28/ICAM-1 for 72 hr. As shown in *Figure 3A,B*, both T-bet and c-Myc did not unevenly segregate into the two nascent daughter cells during telophase. Moreover, the slope of the linear regression curve for the distribution of T-bet and c-Myc as compared to CTV was close to 1, indicating that the repartition of these two molecules in telophase followed that of total proteins.

To further define whether the observed uneven repartition of lytic components was or was not related to ACD, we investigated whether uneven repartition of lytic components was dependent on a polarity cue (e.g. localized TCR stimulation) as previously described for ACD (*Arsenio et al., 2015*; *Pham et al., 2014*). *Figure 4A,B* shows that a polarity cue was not required to induce uneven distribution of lytic molecules, since comparable CD107a[+] vesicle segregation was observed in peripheral blood CD8[+] T cells stimulated by either immobilized (anti-CD3/anti-CD28/ICAM-1) or soluble (phorbol myristate acetate+ ionomycin) stimuli.

Overall, the above results demonstrate that uneven partitioning of lytic compartment in telophase is not associated with asymmetric segregation of fate determining transcription factors. Moreover, a polarity cue is not required. All in all, the above results show that, in human CD8[+] T cells, lytic machinery uneven repartition is not related to described mechanisms of fate determining ACD.

## Asymmetric repartition of CD107a[+] vesicles resets at each division event and generates heterogeneous daughter cells

We next investigated whether lytic machinery uneven repartition occurred during subsequent divisions and whether this process could be involved in preserving lytic machinery heterogeneity within CD8[+] T cell populations.

We considered the cells in the different rounds of division (identified by different peaks of CTV dilution, *Figure 1—figure supplement 3*) and analyzed CD107a repartition in telophasic cells. This analysis showed that, in all division rounds considered, a comparable percentage of cells underwent heterogeneous repartition of CD107a (*Figure 5A,B*).

A complementary observation indicated that the heterogeneity process is stationary, but not hereditary: for example a daughter cell originating from a heterogeneous division has a constant stationary probability to produce a new uneven division. We arrived to this conclusion by generating CD107a fluorescence intensity (CD107a-FI) density curves of all telophasic cells having undergone zero, one, or two mitoses. Cells in telophase showing unequal CD107a-FI repartition were then plotted on these curves (*Figure 5C*). The $\chi^2$ statistical test showed that these cells were randomly and independently distributed on the CD107a-FI density curves, supporting the hypothesis that there is no inheritance in the decision to divide unevenly (see Materials and methods, Table 1).

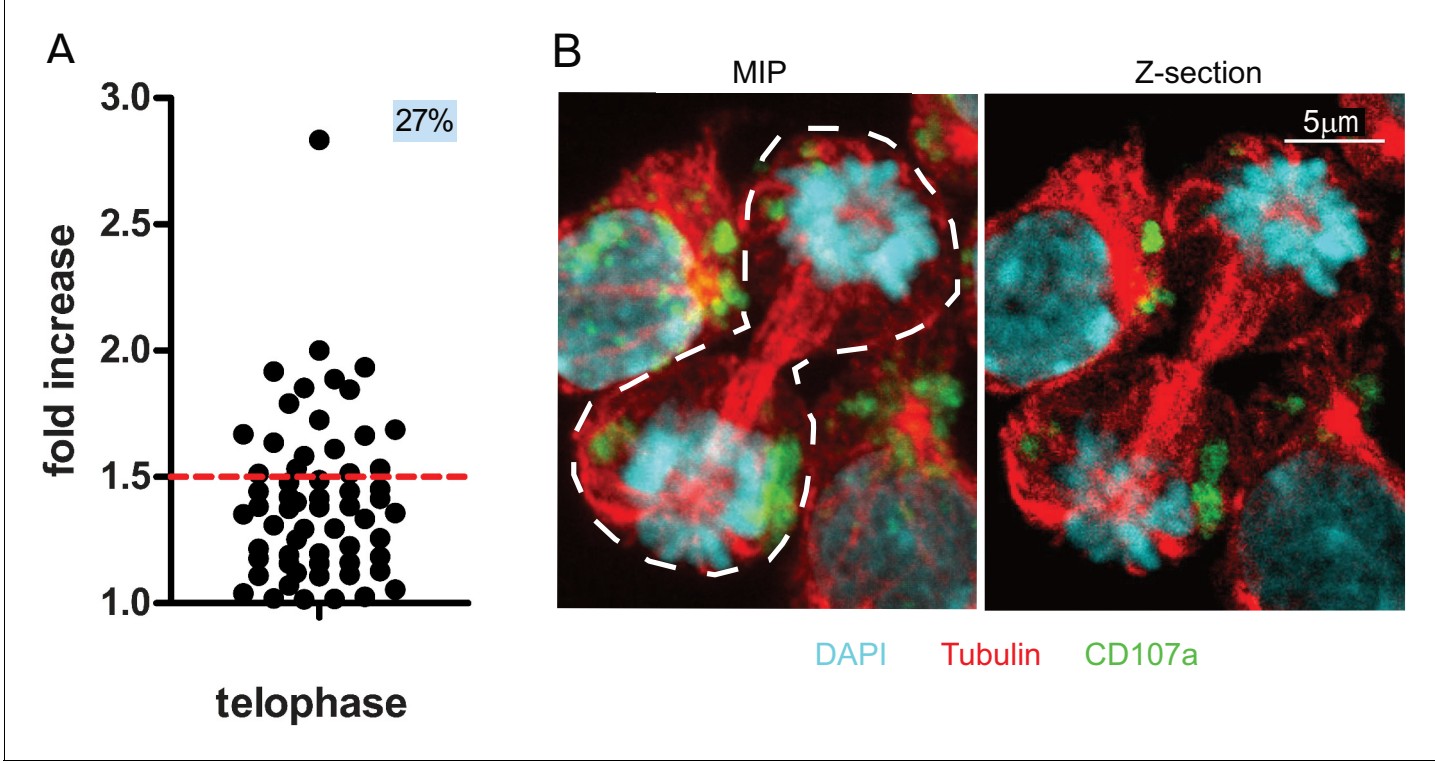

**Figure 2.** CD107a[+] vesicle uneven segregation in telophase is confirmed by confocal laser scanning microscopy. Freshly isolated polyclonal CD8[+] T cells were stimulated by immobilized anti-CD8/anti-CD28/ICAM-1 during 72 hr and stained with antibodies directed against CD107a. Cells in telophase were identified using confocal laser scanning microscopy. (**A**) Analysis of CD107a repartition in dividing cells. The fold increase of CD107a staining in the brighter nascent daughter cell as compared to the other nascent daughter cell is shown. The dotted red line indicates the limit between symmetric and asymmetric cells (1.5 fold increase, corresponding to a 60–40% variation) (n = 61 from two independent experiments). Each dot represents one CD8[+] T cell in telophase. (**B**) Example of an asymmetric cell in division. Green CD107a, cyan DAPI, red Tubulin. A maximum intensity projection (MIP) of a z-stack of images (left panel) and one z-section (right panel) are shown. See *Figure 2—video 1*.

The online version of this article includes the following video and source data for figure 2:

**Source data 1.** Fold increase of CD107a staining in the brighter nascent daughter cell as compared to the other nascent daughter cell.

**Figure 2—video 1.** 3D visualization of CD107a repartition in a telophasic CD8[+] T cell.

https://elifesciences.org/articles/62691#fig2video1

We next asked whether this process might create a drift in lytic compartment content in daughter cells leading to the emergence of cellular subsets expressing higher or lower levels of CD107a. To address this question, we analyzed the total CD107a-FI in all G1 cells (either undivided or following each division round). As shown in *Figure 5D*, the total CD107a-FI appeared to be broadly similar in the different rounds of division in the whole populations, suggesting that uneven repartition of CD107a, in a relatively constant fraction of cells at each division round, does not lead to the emergence of well-defined cellular subsets expressing higher or lower levels of CD107a. We employed the Kolmogorov–Smirnov goodness-of-fit test to determine whether the different curves followed the same distribution or not. The test strongly rejected the hypothesis that the CD107a expression curves follow the same distribution during the first two division rounds (see Materials and methods, Table 2), indicating that during these division events randomly heterogeneous populations were generated. Nevertheless, our test also showed that the Kolmogorov distance decreased when the number of divisions increased, indicating that CD107a-FI density distribution seems to be convergent with a higher number of divisions. To define where variability was located in the curves, we employed the $\chi^2$ test. The test showed that variability was distributed all over the curves (i.e. for all the CD107a-FI). Together, Kolmogorov–Smirnov goodness-of-fit and $\chi^2$ tests revealed a non-stationary variability in the content of CD107a[+] vesicles in CD8[+] T cells during early division events.

Taken together, the above results indicate that asymmetric distribution of CD107a$^+$ vesicles in telophase is not limited to the first division, but it is rather a stochastic process, inherent to each division, that perpetuates variability in daughter cells.

## Lytic granules randomly distribute on the two sides of the cleavage furrow

To gain direct information about the possibility that lytic components might stochastically distribute in nascent daughter cells, we visualized lytic granule repartition during division in individual CTL transfected with mCherry-tagged GrzB mRNA, by live-cell microscopy. mCherry-tagged GrzB showed no preferential localization within cell cytosol at the different phases of the division and appeared to randomly partition into the two nascent daughter cells. In some cases, nascent daughter cells exhibited approximately similar repartition of lytic granules (*Figure 6A*, *Figure 6—video 1*), and in some other cases, lytic granule repartition appeared to be rather asymmetric (*Figure 6B*, *Figure 6—video 2*). Furthermore, we investigated cell division in 4D (3D plus time). Sorted CD8$^+$ T cells in G2/M phase were loaded with LysoTracker Red (LTR) to stain their late endosomal lysosomal vesicles (of which lytic granules are an important fraction [*Faroudi et al., 2003*]). Nascent daughter cells were imaged to monitor the distribution of LTR$^+$ vesicles and measure the integrated fluorescence intensity. An example of one CD8$^+$ T cell distributing LTR$^+$ vesicles in a symmetric fashion during division is shown in *Figure 6C*, *Figure 6—video 3* (LTR distribution ranged within 40–60% at all time points measured). One CD8$^+$ T cell that distributed in an asymmetry fashion LTR$^+$ vesicles is shown in *Figure 6D*, *Figure 6—video 4* (LTR distribution ranged above or below 40–60% at all time points measured). Additional examples of cells dividing in symmetric and asymmetric fashion are shown in *Figure 6—figure supplement 1* and *Figure 6—video 5*.

While lytic granules seemed to be overall randomly distributed between nascent daughter cells, in some cases the videos showed that lytic granules did not behave completely independently from each other and exhibited some clustering. We therefore used a computational approach to establish whether the above-described process might be linked to a random repartition of lytic components into the two nascent daughter cells. We first calculated the probability to obtain an asymmetric distribution of lytic granules (e.g. a repartition of the granules into the two daughter cells out of the 40–60% range) related to the granule number per dividing cell. This computation is naturally handled with a binomial modeling for the behavior of the population of n granules (see Materials and methods). This analysis showed that for n < 100 the probabilities that individual particles distribute asymmetrically on the two sides of the cleavage furrow are relatively high (*Figure 6E*). Using stimulated emission depletion (STED) on CTL stained for GrzB, we estimated that 14–65 (mean = 37) lytic granules are contained within individual CTL. Our estimation well matched with numbers published in previous studies, ranging between 10 and 100 (*Chiang et al., 2017*; *Clark et al., 2003*; *Kataoka et al., 1996*; *Peters et al., 1991*).

These values are compatible with a significant probability of stochastic uneven repartition of lytic granules in telophase, assuming that all lytic granules behave independently.

Since our videos indicate that lytic granules might form transitory aggregates within confined intracellular spaces, we upgraded our mathematical simulation of lytic granule repartition in telophase to include the possibility that lytic granules might not segregate completely independently. We simulated particle correlation during cell division for 10–100 particles. To evaluate the correlation level between individual particles during cell division (ranging from 0 = absence of correlation to 1 = 100% correlation) for a given probability of asymmetric repartition (outside the interval [40%–60%]), we used a Monte-Carlo approach (see Materials and methods).

The analysis shows that for a probability of 20% asymmetric repartition of particles (corresponding to 20% uneven repartition of lytic granules during cell division experimentally measured by imaging flow cytometry and confocal imaging), particle correlation has a relatively low value (4% for 37 particles, 0.04; CI 95%, 0.035–0.045), suggesting that lytic granules mainly segregate independently during cell division.

Taken together, cell imaging and computational results strongly suggest that the observed stationary unequal distribution of lytic granules in telophase is the result of a stochastic repartition of particulate cytosolic structures on the two sides of the cleavage furrow in dividing cells.

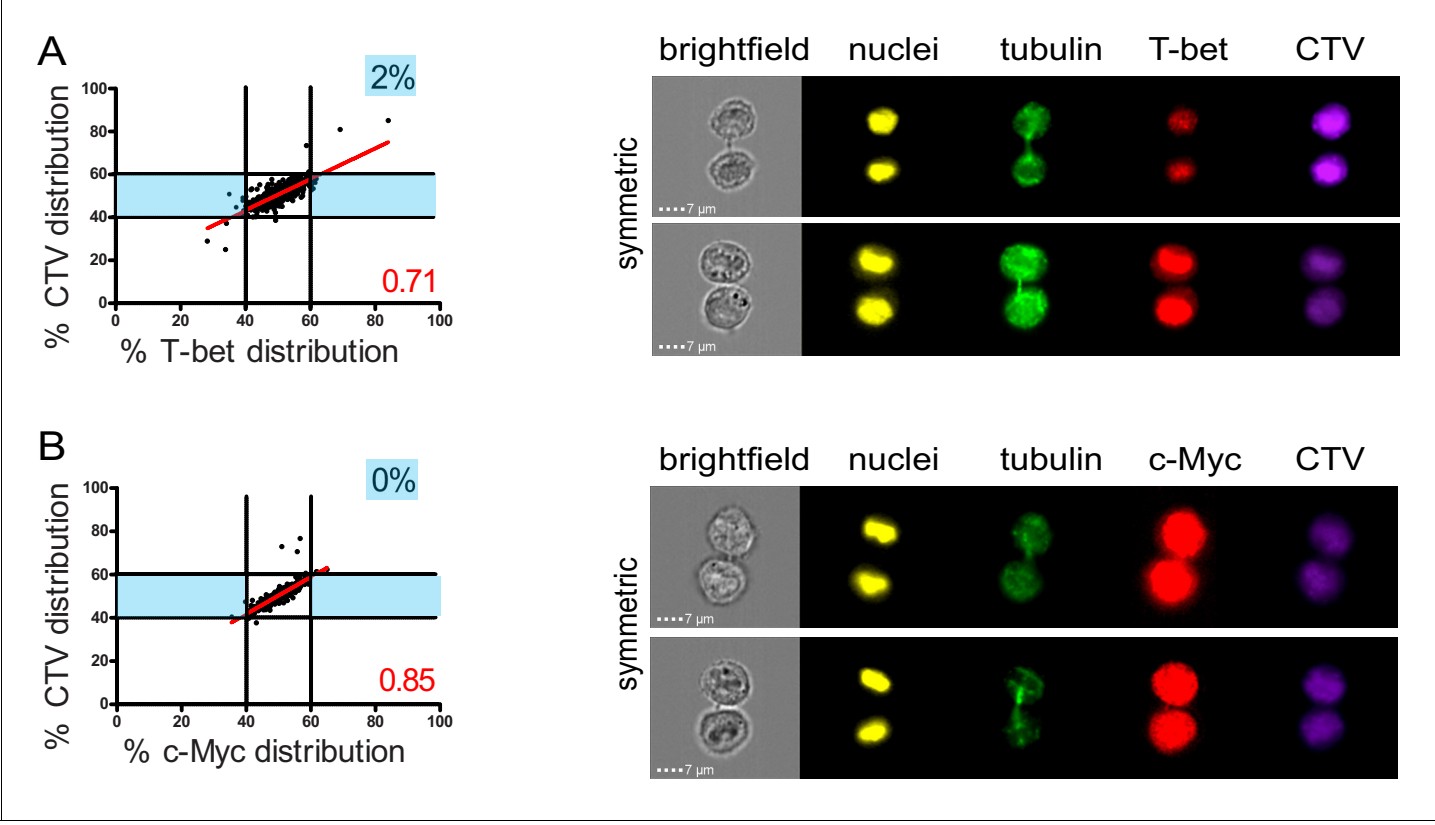

**Figure 3.** Fate determining transcription factors do not undergo uneven distribution in telophase. Freshly isolated polyclonal CD8[+] T cells were stimulated by immobilized anti-CD8/anti-CD28/ICAM-1 during 72 hr and stained with antibodies directed against T-bet (A) or c-Myc (B). (A) T-bet analysis (n = 926 from three independent experiments). (B) c-Myc analysis (n = 703 from three independent experiments). Numbers highlighted in blue in the plots indicate the % of cells exhibiting asymmetric repartition of the marker of interest. Red lines indicate the global distribution of the data. Red numbers indicate the slope of the linear regression curve for marker distribution.

The online version of this article includes the following source data for figure 3:

**Source data 1.** T-bet and c-Myc distribution between daughter cells.

## The level of lytic granule content in individual CTL dictates CTL killing capacity

To assess the consequences of an uneven distribution of lytic compartment on CTL-mediated cytotoxicity, we investigated cytotoxic efficacy in CTL expressing high and low lytic granule content. Clonal CTL were loaded with LysoTraker blue, and cells containing high (LysoTracker[High]) and low (LysoTracker[Low]) levels were sorted by fluorescence-activated cell sorting. As shown in *Figure 7A*, sorted LysoTracker[High] and LysoTracker[Low] CTL populations maintained their difference in Lyso-Tracker staining at least 24 hr after cell sorting. The cytotoxic efficacy of sorted CTL populations was compared at different effector:target (E:T) ratios by measuring the percentage of killed targets (7-aminoactinomycin D [7-AAD]-positive targets). For each ratio, LysoTracker[High] CTL were more efficient than LysoTracker[Low] CTL in exerting cytotoxicity (*Figure 7B,C*), although basal killing (in the absence of peptide stimulation) was comparable between LysoTracker[High] and LysoTracker[Low] CTL (*Figure 7C*). The above results indicated that lytic granule content is associated with killing efficacy. To strengthen these findings, we performed additional experiments on sorted CTL for high and low LysoTracker staining and measured CD107a surface exposure and CD8 internalization following 4 hr conjugation with target cells. Results show that LysoTracker[high] CTL exhibited a higher lytic granule secretion as detected by CD107a exposure when compared to LysoTracker[low] CTL (*Figure 7D*). However, productive TCR engagement was comparable in both populations as detected by similar levels of CD8 internalization (*Huang et al., 2019*; *Xiao et al., 2007*; *Figure 7E*).

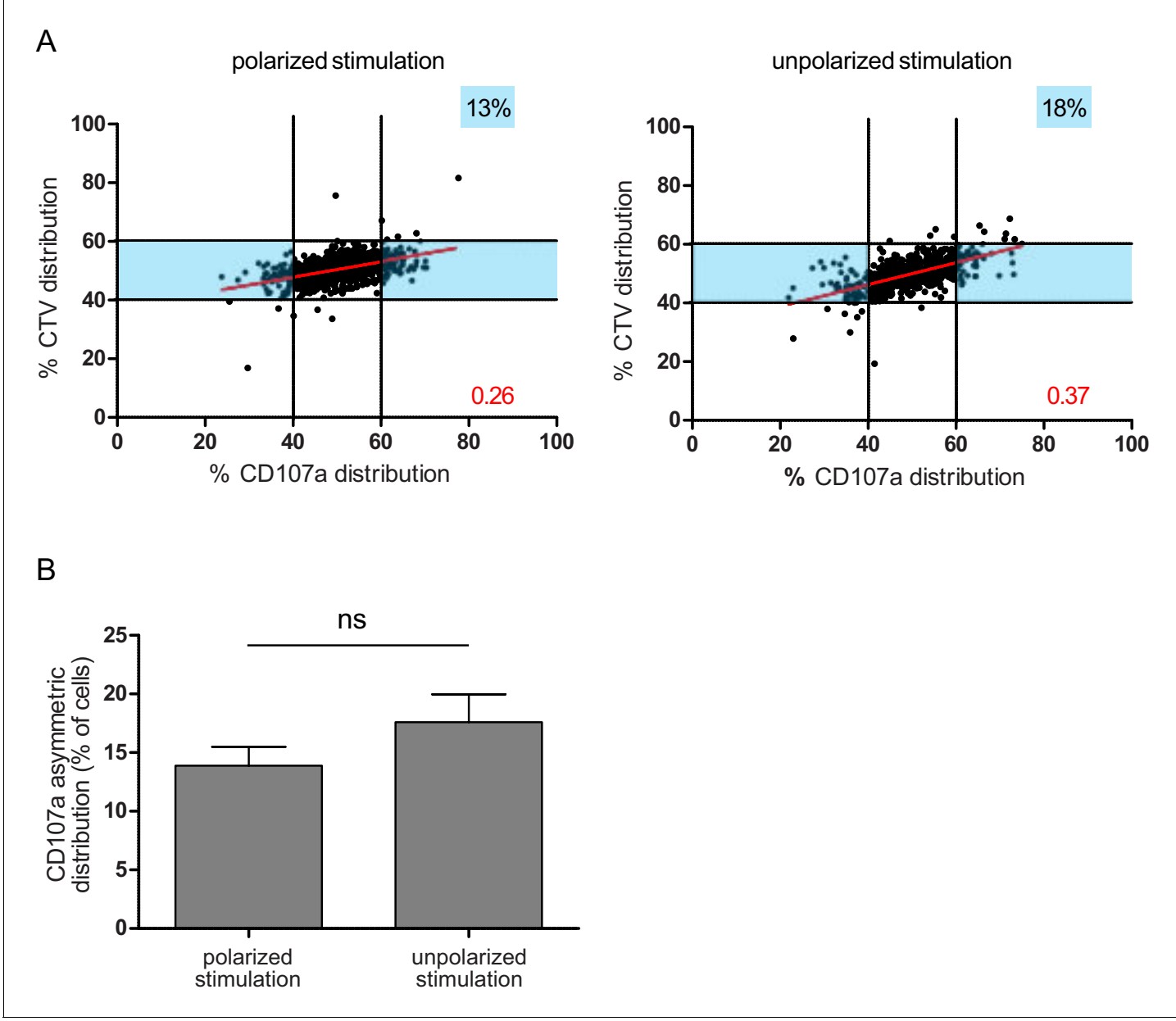

**Figure 4.** A polarity cue is not necessary for asymmetric repartition of lytic machinery. (**A**) Freshly isolated polyclonal CD8[+] T cells were stimulated using immobilized anti-CD8/anti-CD28/ICAM-1 (left) or with PMA/ionomycin (right) during 72 hr and stained with antibodies directed against CD107a. Each dot represents one nascent daughter cell. Only one of the two nascent daughter cells in telophase that were identified by Imaging Flow Cytometry is plotted. The percentage of staining for CD107a in the presented nascent daughter cell (x axis) is plotted against the percentage of staining for total cell proteins (CTV, y axis). Asymmetric cells were defined as in *Figure 1*. Left: CD107a analysis when cells were stimulated with immobilized stimuli (n = 1185 from three independent experiments). Right: CD107a analysis when cells were stimulated with PMA/ionomycin (n = 644 from three independent experiments). Numbers highlighted in blue in the plots indicate the % of cells exhibiting asymmetric repartition of the marker of interest. Red lines indicate the global distribution of the data. Red numbers indicate the slope of the linear regression curve for CD107a distribution. (**B**) Histograms represent the mean and standard deviation of the percentage of asymmetric cells in the three independent experiments. No statistical difference was revealed by paired t-test.

The online version of this article includes the following source data for figure 4:

**Source data 1.** CD107a distribution between daughter cells after polarized and non-polarized stimulation.

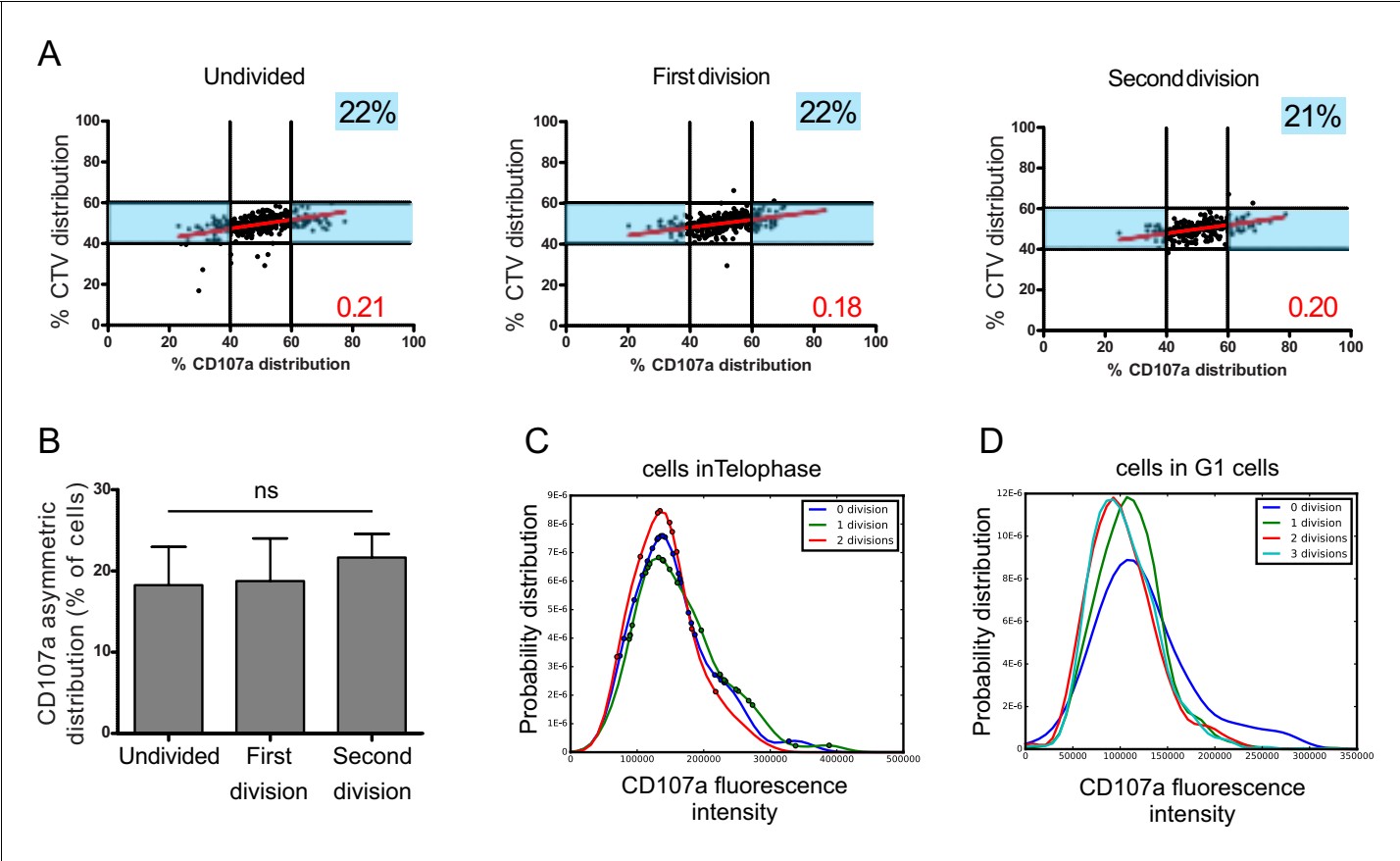

**Figure 5.** Asymmetric repartition of CD107a⁺ vesicles resets at each division event. (**A, B**) Freshly isolated polyclonal CD8⁺ T cells were stimulated using immobilized anti-CD8/anti-CD28/ICAM-1 during 72 hr and stained with antibodies directed against CD107a. Cells in telophase were identified by imaging flow cytometry. The number of divisions accomplished and the cell cycle phase were determined on the basis of CTV and SYTOX nuclear staining. (**A**) Each dot represents one nascent daughter cell. Only one of the two nascent daughter cells in telophase that were identified by imaging flow cytometry is plotted. The percentage of staining for CD107a in the presented nascent daughter cell (x axis) is plotted against the percentage of staining for total cell proteins (CTV, y axis). Asymmetric cells were defined as in *Figure 1*. Numbers highlighted in blue in the plots indicate the % of cells exhibiting asymmetric repartition of the marker of interest. Red lines indicate the global distribution of the data. Red numbers indicate the slope of the linear regression curve for CD107a distribution. See Figure S3. (**B**) Histograms represent the mean and standard deviation of the percentage of asymmetric cells in three independent experiments. No statistical difference was revealed by paired t-test. (**C, D**) Statistical analysis of cells in telophase and in G1. (**C**) Cells in telophase are plotted against their CD107a FI. The different curves represent cells having undergone zero, one, or two mitoses. Each dot indicates one cell undergoing asymmetric CD107a repartition as compared to its CD107a FI. The $\chi^2$ statistical test showed that cells undergoing uneven repartition of lytic machinery in telophase were randomly distributed all over the CD107a expression curves (See Materials and methods). (**D**) Plots show cells in G1 from three different experiments. Curves represent the distribution of CD107a florescence intensity for all cells in G1. Individual plots, marked with different colors, show cells in G1 at different rounds of division. The Kolmogorov–Smirnov goodness-of-fit test rejected the hypothesis that the CD107a expression curves follow the same distribution at the different division round (see Supplementary Results). The $\chi^2$ test showed that variability was distributed all over the curves. See Figure S3.

The online version of this article includes the following source data for figure 5:

**Source data 1.** CD107a distribution between daughter cells in undivided cells, at first division and second division.

Together, these results indicate that the lytic granule cargo of individual CTL and not their activation properties directly impact killing behavior. They imply that stochastic uneven distribution of lytic granules in dividing CTL determine heterogeneous killing behavior at the single-cell level.

## Discussion

In the present study, we found that, in both freshly isolated peripheral blood CD8⁺ T cells and clonal CTL, ~20% of telophasic cells undergo asymmetric distribution of the lytic compartment into the two daughter cells. Our results establish that CD8⁺ killing capacity is associated with lytic compartment

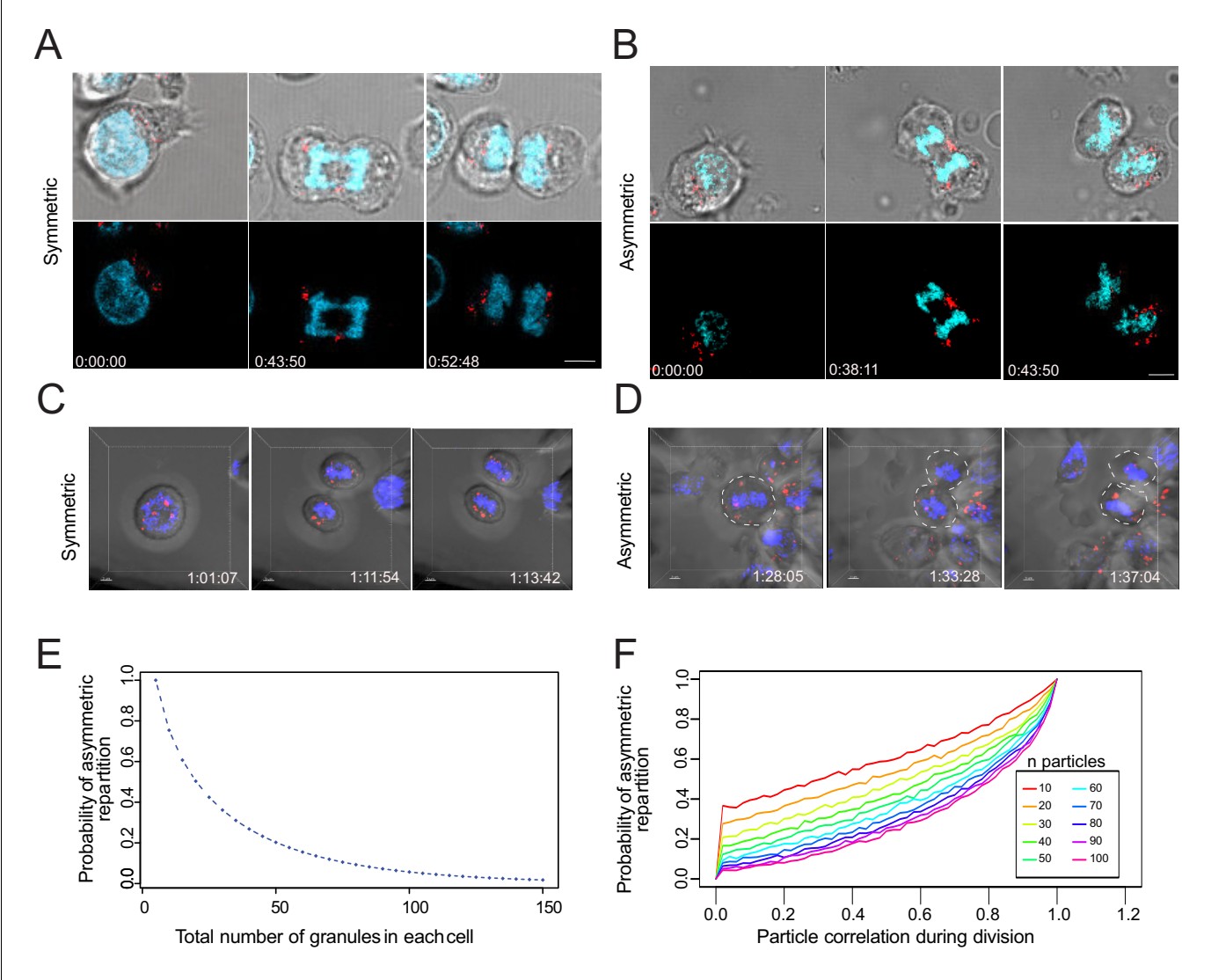

**Figure 6.** Lytic granules randomly distribute on the two sides of the cleavage furrow. (A,B) Snapshots depict typical cells in division undergoing even (A) or uneven (B) repartition of lytic granules (mCherry-tagged GrzB, red) in telophase as detected by live-cell imaging. Images are from *Figure 6—videos 1* and *2*, respectively. Results are from three independent experiments. (C,D) Snapshots depict Imaris software reconstructions of typical cells undergoing even (C) or uneven (D) repartition of LTR+ (red) lytic granules in division as detected by 4D live-cell imaging. Images are from *Figure 6—videos 3* and *4*, respectively. Results are from four independent experiments. See *Figure 6—videos 3–5*. (E) Binomial modeling for the behavior of the population of n granules. The curve shows the probability of lytic granule asymmetric repartition in telophase as a function of lytic granule number. (F) Monte-Carlo simulation of particle correlation as a function of lytic granule number and probability of lytic granule asymmetric repartition.
The online version of this article includes the following video and figure supplement(s) for figure 6:

**Figure supplement 1.** Lysotracker randomly distribute on the two sides of the cleavage furrow.
**Figure 6—video 1.** Symmetric repartition of granzyme B during cell division.
https://elifesciences.org/articles/62691#fig6video1
**Figure 6—video 2.** Asymmetric repartition of granzyme B during cell division.
https://elifesciences.org/articles/62691#fig6video2
**Figure 6—video 3.** Symmetric repartition of LTR+ vesicles during cell division.
https://elifesciences.org/articles/62691#fig6video3
**Figure 6—video 4.** Asymmetric repartition of LTR+ vesicles during cell division.
https://elifesciences.org/articles/62691#fig6video4
**Figure 6—video 5.** LTR+ vesicles repartition during cell divisions.
https://elifesciences.org/articles/62691#fig6video5

level and strongly suggest that uneven lytic machinery repartition produces CD8$^+$ T cell populations with heterogeneous killing capacities.

We used imaging flow cytometry, a technique that combines the advantages of flow cytometry and microscopy and allows the detection and analysis of rare cells within whole-cell populations on the basis of their morphological and staining characteristics (*Basiji and O'Gorman, 2015*; *Doan et al., 2018*; *Hritzo et al., 2018*). We thus acquired and analyzed a significant number of relatively rare events of T cell divisions by precisely identifying cells in telophase. The use of CTV distribution as a parameter of global protein repartition in telophase, together with the acquisition of an important number of cell divisions, strengthens our analysis. In addition, we investigated lytic granule repartition in dividing CD8$^+$ T cells by 3D confocal laser scanning microscopy and 4D live-cell imaging. These techniques allowed visualization of lytic granule repartition in telophase with a high time/space resolution and strengthened imaging flow cytometry data by providing unambiguous visualization of lytic granule partitioning.

Our results demonstrate that the uneven lytic machinery distribution is not related to ACD.

In mouse T lymphocytes, ACD has been reported as a mechanism contributing to the generation of effector/memory daughter cells following the division of an individual naive T cell in response to polarizing cues (*Arsenio et al., 2015*; *Chang et al., 2007*). Establishment of asymmetry has been associated with the uneven inheritance by daughter cells of transcription factors such as c-Myc and T-bet known for their role in the induction of metabolic reprogramming and in the acquisition of T cell effector function, respectively (*Chang et al., 2011*; *Verbist et al., 2016*). Following the original observation of uneven repartition of proteasomes in dividing mouse CD4$^+$ T cells leading to asymmetric degradation of T-bet in daughter cells (*Chang et al., 2011*), additional cellular effectors including metabolic and signaling pathways have been found to be implicated in fate determining ACD in mouse naive T lymphocytes (*Kamiński et al., 2016*; *Pollizzi et al., 2016*; *Verbist et al., 2016*). Our results, by showing that lytic granule repartition is not accompanied by a detectable asymmetric segregation of T-bet and c-Myc and does not require a polarity cue, suggest that the lytic machinery uneven distribution observed in human CD8$^+$ T cells is not related to previously described ACD. Although we could not detect an asymmetric repartition of classical lineage-determining transcription factor, in our models, this observation does not exclude the possibility that ACD might play a role in the differentiation of human naive T cells into effector and memory subsets during initial antigen-specific immune responses. It is therefore possible that the discrepancy between our results and previous studies on ACD in mouse T lymphocytes arises from the different nature of the cells involved in the study. It should also be noted that, besides ACD, other mechanisms can contribute to the generation of different T lymphocyte populations from naive lymphocytes and, more in general, can play a role in T lymphocyte heterogeneity. Alternative models postulate that lymphocyte differentiation might be achieved via the accumulation of progressive differences among daughter cells due to variation in the quantity of the inherited proteins (*Buchholz et al., 2016*; *Cobbold et al., 2018*; *Gerlach et al., 2013*; *Girel et al., 2019*; *Pham et al., 2014*; *Rohr et al., 2014*; *Schumacher et al., 2010*).

A puzzling question is how asymmetric distribution of lytic components in telophase is generated. Our results provide a stepping-stone to answer this question. First, mathematical analysis of our imaging flow cytometry data provides an interpretation of our results that is compatible with a stochastic distribution of lytic components during cell division. On the one hand, mathematical analysis shows that the process of asymmetric distribution is stationary in terms of the fraction of involved cells: for example occurs always on a similar percentage of cells, at each division round, in different experiments and following different stimuli. On the other hand, the heterogeneity process, although stationary, is not hereditary: for example a daughter cell originating from a heterogeneous division has a constant stationary probability to produce a new uneven division. Second, live-cell imaging shows lytic granule distribution during mitosis. We did not observe any specific pattern of lytic granule repartition (polarization at the membrane or close to the cleavage furrow) before or during cell division. Instead, lytic compartments appeared randomly distributed in cell cytosol. Our observations are consistent with the mathematical modeling of intracellular vesicle distribution showing the high probability of an uneven distribution of a relatively small quantity of granules. In other words, prepackaged molecular components within a few relatively big vesicles might have higher probability to be asymmetrically partitioned in telophase than molecular components dispersed throughout the cytosol.

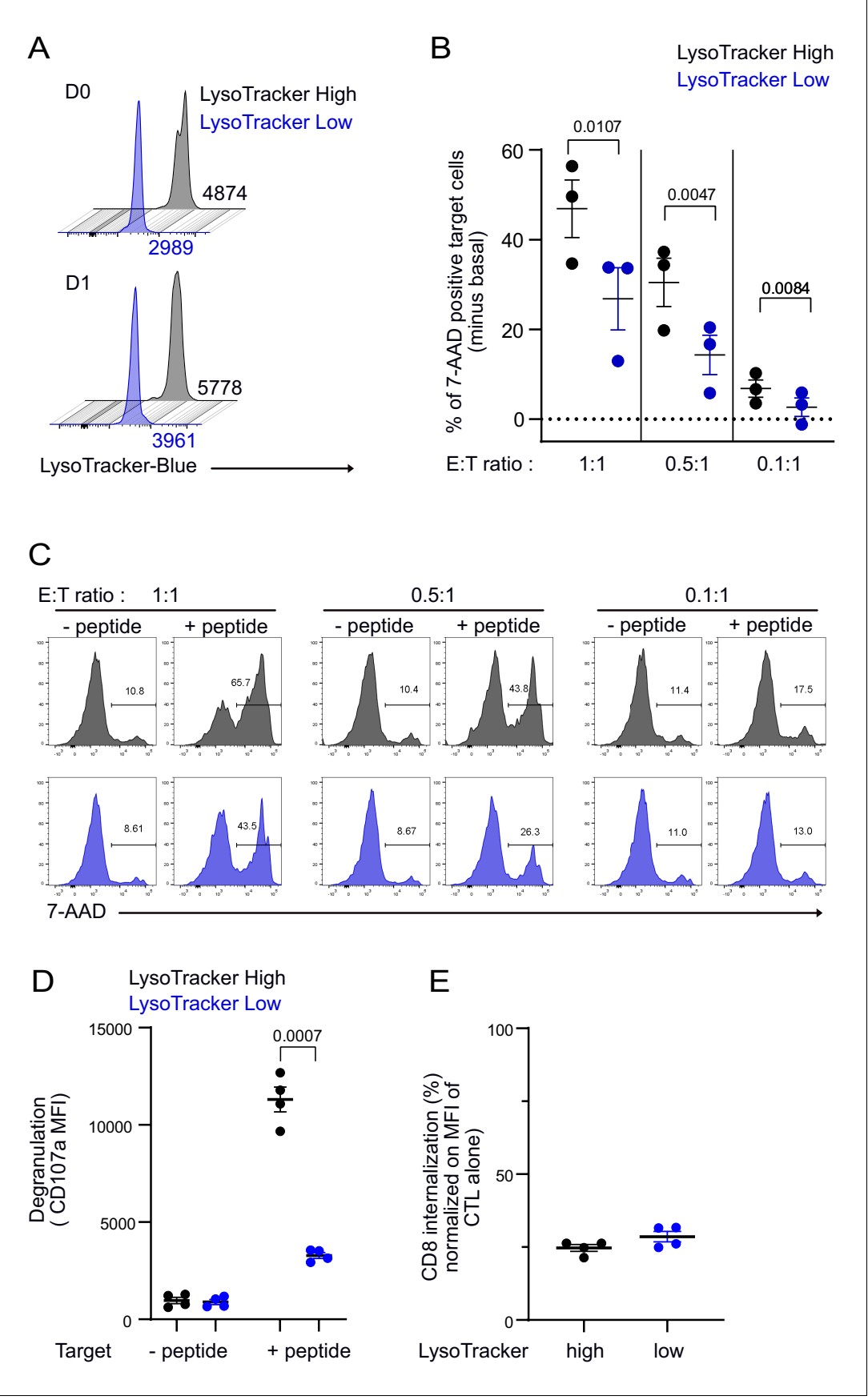

**Figure 7.** CTL expressing high level of lytic granules have better killing capability. Clonal CTL were FACS sorted on the basis of their LysoTracker Blue staining. (**A**) Representative FACS histograms showing LysoTracker Blue staining levels on LysoTracker[high] and LysoTracker[low] sorted CTL at the indicated day (**D**) after cell sorting. Numbers indicate mean fluorescence intensity. Results are representative of three independent experiments (**B,C**) LysoTracker[High] and LysoTracker[Low] CTL-mediated cytotoxicity was evaluated by FACS analysis by measuring 7-AAD uptake in target cells either pulsed or not with antigenic peptide following overnight incubation with CTL at the indicated E/T ratio. (**B**) Cytotoxicity is expressed as the % of 7-AAD[+]-pulsed target cells minus % of 7-AAD[+]-unpulsed target cells (basal). Results are from three independent experiments. Each dot represents results from one experiment performed in triplicate. Means ± SEM are shown. Paired t-tests were performed, and p-values are indicated. (**C**) Histograms shown are from one representative experiment. Numbers indicate the percentage of 7-AAD-positive target cells. (**D**) LysoTracker[High] and LysoTracker[Low] CTL CD107a exposure after a 4 hr incubation with target cells pulsed or not with antigenic peptide (E/T ratio 0.5:1) was evaluated by FACS analysis. Each dot represents results from four independent experiments performed either in duplicate or triplicate. Means ± SEM are shown. Paired t-tests were performed, and p-values are indicated. (**E**) CD8 expression in LysoTracker[High] and LysoTracker[Low] CTL after a 4 hr incubation with target cells pulsed with antigenic peptide (E/T ratio: 0.5:1) was evaluated by FACS analysis. Results are normalized on CD8 MFI level of LysoTracker[High] and LysoTracker[Low] CTL cultured in the absence of target cells. Each dot represents results from four independent experiments performed either in duplicate or triplicate. Means ± SEM are shown.

The online version of this article includes the following source data for figure 7:

**Source data 1.** Cytotoxicity assay.
**Source data 2.** CD107a mean fluorescence intensity (MFI) at the surface of CD8[+] T cells.
**Source data 3.** CD8 mean fluorescence intensity (MFI).

---

Moreover, it should be noted that our videos and results in *Figure 6* suggest that, in a limited number of division events, granules might not segregate completely independently from each other, implying that some active mechanism of granule segregation might contribute, to a minor extent, to lytic granule repartition in telophase.

Together, our results point out a mechanism of heterogeneity generation that is for the most part stochastic and might be a general mechanism for generating heterogeneity in dividing cells. The possibility that particulate material is unevenly distributed in telophase into two nascent daughter cells has been proposed for other organelles and in other cellular systems (*Bergeland et al., 2001*; *Carlton et al., 2020*; *Sanghavi et al., 2018*). Indeed, in MDCK cells, microscopy analysis and mathematical modeling based on the laws of probability suggested that endosome/lysosome partitioning between daughter cells is stochastic (*Bergeland et al., 2001*). Others show that in telophasic cells, endosomal compartments are clustered at the cleavage furrow, suggesting that microtubules are involved in this process. However, no mechanism ensuring endosomal compartment anchorage to either spindle has been revealed, suggesting that this repartition is stochastic. Similarly, in Dictyostelium cells, it has been demonstrated that dynein and kinesin motors drive phagosomes segregation independently of each other and stochastically (*Sanghavi et al., 2018*). To our knowledge, our present study is the first to relate a mechanism of a random segregation of organelles to functional heterogeneity of immune cells.

What could be the functional role of asymmetric molecular segregation during mitosis in human CD8[+] T cells? We propose that a mechanism of asymmetric distribution in telophase (that is stationary at each division, but not inherited by daughter cells) can be instrumental to randomly generate short-lived CTL cohorts harboring functional heterogeneity while ensuring globally reproducible antigen-specific CD8[+] T cell responses. This process might confer robustness to CTL responses through population averaging (*Buchholz et al., 2016*; *Hodgkin et al., 2014*).

It is interesting to note that our results present analogies with previously published data in which asymmetric segregation of internalized exogenous antigen was found to occur during B cell division (*Thaunat et al., 2012*). Together with this previous study, our results reveal an intriguing capacity of both T and B cells to stochastically distribute in telophase their acidic compartments: MHC Class II compartments for B cells and lytic granules for CD8[+] T cells. Thus, stochastic distribution in telophase appears to be a major mechanism ensuring a high variability of both humoral and cellular adaptive immune responses during lymphocyte clonal expansion.

# Materials and methods

## Key resources table

| Reagent type (species) or resource | Designation | Source or reference | Identifiers | Additional information |
|---|---|---|---|---|
| Cell line (*Homo sapiens*) | HLA-A2 restricted CD8+ T cell clone (VLAELVKQI) | *Khazen et al., 2016* | | |
| Cell line (*Homo sapiens*) | HLA-A2 restricted CD8+ T cell clone (NLVPMVATV) | *Khazen et al., 2016* | | |
| Cell line (*Homo sapiens*) | HLA-A2 restricted CD8+ T cell clone (VLAELVKQI) | *Khazen et al., 2016* | | |
| Cell line (*Homo sapiens*) | JY (EBV-transformed B cells) | *Khazen et al., 2016*; *Vasconcelos et al., 2015* | | |
| Biological sample (*Homo sapiens*) | Buffy coats of Healthy donors | EFS, Toulouse, France | | With consent and approval AC-2014–2384 |
| Antibody | Anti-human CD3 (Human monoclonal, TR66) | Enzo | Cat# ALX-804–822 RRID:AB_2051037 | (1 µg/ml) |
| Antibody | Anti-human CD28 (Mouse monoclonal, CD28.2) | eBioscience | Cat# 16-0289-81 RRID:AB_468926 | (1 µg/ml) |
| Recombinant protein | Recombinant human ICAM-1-Fc fusion protein | R and D Systems | Cat# 720-IC | (0.5 µg/ml) |
| Antibody | Anti-human CD107a (Mouse monoclonal, H4A3) | BD Pharmingen | Cat# 555798 RRID:AB_396132 | (10 µg/ml) |
| Antibody | Anti-human CD107a AlexaFluor 647 (Mouse monoclonal, H4A3) | BD Pharmingen | Cat# 562622 RRID:AB_2737684 | (Diluted at 1/100) |
| Antibody | Anti-human Granzyme B (Mouse monoclonal, GB11) | Thermo Scientific | Cat# MA1-80734 RRID:AB_931084 | (10 µg/ml) |
| Antibody | Anti-human Granzyme B AlexaFluor 647 (Mouse monoclonal, GB11) | BD Pharmingen | Cat# 561999 RRID:AB_10897997 | (10 µg/ml) |
| Antibody | Anti-human T-bet (Rabbit polyclonal, Tbx21) | Abcam | Cat# ab181400 | (10 µg/ml) |
| Antibody | Anti-human C-myc (Mouse monoclonal, 9E10) | Thermo Scientific | Cat# MA1-980 RRID:AB_558470 | (10 µg/ml) |
| Antibody | Anti-human α-tubulin (Rabbit polyclonal) | Abcam | Cat# ab15246 RRID:AB_301787 | (Diluted at 1/100) |
| Antibody | Anti-mouse IgG1 Alexa Fluor 647 (Goat polyclonal) | Invitrogen | Cat# A21240 RRID:AB_2535809 | (10 µg/ml) |
| Antibody | Anti-mouse IgG1 Alexa Fluor 488 (Goat polyclonal) | Invitrogen | Cat# A-21121 RRID:AB_2535764 | (10 µg/ml) |
| Antibody | Anti-rabbit (H+L) AlexaFluor488 (Goat polyclonal) | Invitrogen | Cat# A11034 RRID:AB_2576217 | (10 µg/ml) |
| Antibody | Anti-rabbit (H+L) AlexaFluor647 (Donkey polyclonal) | Invitrogen | Cat# A31573 RRID:AB_2536183 | (10 µg/ml) |
| Antibody | Anti-rabbit AlexaFluor555 (Goat polyclonal) | Invitrogen | Cat# A21428 RRID:AB_2535849 | (10 µg/ml) |
| Antibody | Anti-mouse IgG Abberior Star 580 (Goat polyclonal) | Abberior Instruments | Cat# 52403 | (10 µg/ml) |
| Antibody | Anti-human CD107a-PEcy7 (mouse monoclonal, H4A3) | BD Pharmingen | Cat# 561348 RRID:AB_10644018 | (Diluted at 1/50) |
| Antibody | Anti-human CD8-FITC (mouse monoclonal, HIT8A) | BD Pharmingen | Cat# 555634 RRID:AB_395996 | (Diluted at 1/50) |

*Continued on next page*

*Continued*

| Reagent type (species) or resource | Designation | Source or reference | Identifiers | Additional information |
|---|---|---|---|---|
| Recombinant DNA reagent | MGC Human GZMB Sequence verified cDNA (Clone Id: 5223876) | GE Healthcare BIO Sciences | Cat# MHS6278-202801737 | |
| Recombinant DNA reagent | mCherry-SEpHluorin | *Koivusalo et al., 2010* | Addgene cat# 32001 | |
| Recombinant DNA reagent | pT7-GZMB-mCherry-SEpHluorin | This paper | | |
| Sequence-based reagent | Primer: XhoI-T7-GzB Forward caaCTCGAGTAATACGACTCACTATAGGGAGACCCGGTACCatgcaaccaatcctgcttctgcc | This paper | | |
| Sequence-based reagent | Primer: EcoRI-GzB-noSTOP-R caaGAATTCcggcgtggcgtttcatggttttctttatccag | This paper | | |
| Peptide, recombinant protein | CMV peptide p65 (NV-9) | GeneCust | Cat# 181329 | |
| Peptide, recombinant protein | Human rIL-2 | Miltenyi Biotec | Cat# 130-097-748 | (150 IU/ml) |
| Peptide, recombinant protein | Human rIL-15 | Miltenyi Biotec | Cat# 130-095-766 | (50 ng/mL) |
| Commercial assay or kit | EasySep Negative human $CD8^+$ T cell isolation kit | StemCell Technologies | Cat# 17953 | |
| Commercial assay or kit | EasySep human Memory $CD8^+$ T cell enrichment kit | StemCell Technologies | Cat# 19159 | |
| Software, algorithm | IDEAS SpotCount Threshold (M03,nucleus,60) | Amnis, Luminex | | |
| Software, algorithm | IDEAS Area Range Threshold (M02,tubulin,75), 50–5000, 0–0.5 | Amnis, Luminex | | |
| Software, algorithm | Fiji | *Schindelin et al., 2012* | | |
| Software, algorithm | Imaris Software | Oxford Instruments | | |
| Software, algorithm | ZEN ZEISS Efficient Navigation | | | |
| Software, algorithm | Huygens Professional version 18.10 using CMLE algorithm with SNR:7 | Scientific Volume Imaging, USA | | STED images were deconvolved |
| Software, algorithm | Python software version 3.5 | | | $\chi^2$ of independence test, $\chi^2$ of homogeneity test and Kolmogorov-Smirnov goodness-of-fit test |
| Software, algorithm | GraphPad Prism software version five for windows | | | Paired Student's t-test |
| Software, algorithm | FlowJo software | TreeStar | | |
| Other | SYTOX Orange Dead Cell Stain | ThermoFisher Scientific | Cat# S11368 | Manufacturer recommended dilution |
| Other | DAPI | Molecular Probes, Invitrogen | Cat# D1306 RRID:AB_2629482 | |
| Other | Hoechst 33342 | ThermoFisher Scientific | Cat# 1399 | (200 ng/ml) |
| Other | CellTrace Violet Cell Proliferation kit | ThermoFisher Scientific | Cat# C34557 | (5 µM) |
| Other | LysoTraker Blue (DND22) Dye | Molecular probes | Cat# L7525 | (200 nM) |
| Other | LysoTraker Red (DND99) Dye | Molecular probes | Cat#L7528 | (200 nM) |

*Continued on next page*

*Continued*

| Reagent type (species) or resource | Designation | Source or reference | Identifiers | Additional information |
|---|---|---|---|---|
| Other | 7-Aminoactinomycin D (7-AAD) | BD Pharmingen | Cat# 559925 | (0.25 µg) |
| Other | Ibidi µ-slide chambered coverslips Angiogenesis | Ibidi, Biovalley | Cat# 81506 | |
| Other | Ibidi µ-slide chambered coverslips eight well | Ibidi, Biovalley | Cat# 80821 | |
| Other | Nunc Lab-Tek chamber slides eight wells | Nunc, ThermoFisher Scientific | Cat#1 54526 | |
| Other | Micromesh array (100 µm) | Microsurface, Tebu- Bio | Cat# MMA-0500-100-08-01 | |

## Experimental model and subject details

Peripheral blood mononuclear cells were isolated from buffy coats of healthy donors obtained through the Etablissement Français du Sang (EFS, Toulouse, France). Blood samples were collected and processed following standard ethical procedures (Helsinki protocol), after obtaining written informed consent from each donor and approval by the French Ministry of the Research (transfer agreement AC-2014–2384). Approbation by the ethical department of the French Ministry of the Research for the preparation and conservation of cell lines and clones starting from healthy donor human blood samples has been obtained (authorization no DC-2018-3223).

## Cell culture and stimulating conditions

Total human CD8$^+$ T cells were purified from healthy donor blood samples using the EasySep Negative Human CD8$^+$ T cell Isolation Kit (StemCell Technologies). CD8$^+$ T cells were routinely ~90% pure. Memory human CD8$^+$ T cells were purified from healthy donor blood samples using the Easy-Sep Human Memory CD8$^+$ T cell Enrichment Kit (StemCell Technologies), and cells were routinely ~90% CD8$^+$RO$^+$RA$^-$.

HLA-A2 restricted CD8$^+$ T cell clones, specific for the NLVPMVATV peptide or the VLAELVKQI peptide of the CMV protein pp65, were cultured in complete RPMI/HS medium (RPMI 1640 medium supplemented with 5% human AB serum; Inst. Biotechnologies J. Boy, Reims), minimum essential amino acids, HEPES, sodium pyruvate (Invitrogen), 2-mercaptoethanol (5 µM, Gibco), and 150 IU/ml human rIL-2 and 50 ng/ml rIL-15. Clones were re-stimulated every 2–3 weeks in 24-well plate with $1 \times 10^6$ irradiated (35 Gy) allogeneic peripheral blood mononuclear cells (isolated on Ficoll Paque Gradient from fresh heparinized blood samples of healthy donors, obtained from EFS) and $1 \times 10^5$ irradiated EBV-transformed B cells. Complete RPMI/HS medium was supplemented with 1 µg/ml PHA.

EBV-transformed B cells (JY) HLA-A2$^+$ were used as target cells and cultured in RPMI 1640 Gluta-MAX supplemented with 10% fetal calf serum (FCS) and 50 µM 2-mercaptoethanol, 10 mM HEPES, 1× MEM non-essential amino acids, 1× sodium pyruvate, and 10 µg/ml ciprofloxacine. Profiling of JY cells has been done using STR.

All cell lines are routinely screened for mycoplasma contamination using MycoAlert mycoplasma detection kit (Lonza, Basel, Switzerland).

For imaging flow cytometry (ImageStreamX, Merk) and confocal laser scanning microscopy human CD8$^+$ T cells or CD8$^+$ T cell clones were stimulated for 72h with immobilized anti-CD3 (1 µg/ml, TR66 [*Valitutti et al., 1995*]), anti-CD28 (1 µg/ml, clone CD28.2, eBioscience), and immobilized recombinant ICAM1-Fc fusion protein (0.5 µg/ml, R&D Systems) in complete RPMI/HS medium. For confocal laser scanning, cells were plated on anti-CD3/CD28/ICAM1 coated Nunc Lab-Tek Chamber Slide system 8 wells at 500,000 cells/well. For image stream analysis, cells were plated on anti-CD3/CD28/ICAM1-coated 24-well plates at $1.5 \times 10^6$ cells/well.

## Image stream analysis
### Staining and acquisition strategy
Cells were first stained with CellTrace Violet Cell Proliferation Kit (CTV) in phosphate-buffered saline (PBS) (5 µM, 20 min, 37°C). After 72 hr of stimulation (cf Cell culture and stimulating condition), cells were fixed in 1% paraformaldehyde (PFA) (10 min, 37°C) and permeabilized in permeabilization buffer (PBS 3% bovine serum albumin [BSA], 0.1% saponin, Sigma) for 5 min. Cells were incubated for 45 min with the indicated antibodies: AlexaFluor 647 anti-human CD107a antibody (diluted at 1/100, clone H4A3; BD Pharmingen ), anti-human Perforin (10 µg/ml, clone δG9; BD Pharmingen ), AlexaFluor 647 anti-human Granzyme B antibody (10 µg/ml, clone GB11, BD Pharmingen ), anti-human T-bet (Tbx21) (10 µg/ml, clone 4B10; Abcam), anti-human C-myc (10 µg/ml, clone 9E10; Thermo scientific), and anti-human α-tubulin (diluted at 1/100, rabbit polyclonal; Abcam). The following secondary antibodies were used: AlexaFluor488 or 647 goat anti-mouse IgG1 (10 µg/ml; Invitrogen), AlexaFluor488 or 647 anti-rabbit (H+L) (10 µg/ml; Invitrogen). For image acquisition, cells were adjusted to $10 \times 10^6$–$20 \times 10^6$ /ml in FACS buffer (PBS, 1% FCS, 5% Hepes, 0.1% Azide) containing SYTOX Orange Dead Cell Stain (recommended dilution, ThermoFisher Scientific) for nuclear staining. Cells were acquired using ImageStreamX (IsX; Amnis, Luminex) technology.

### Telophase discrimination strategy
Amnis IDEAS software was used to analyze IsX data and identify cells in telophase. As in classical cytometry data analysis, cells in G2/M phase were first selected according to their DNA content (fluorescence of SYTOX orange). A mask based on nuclear staining was employed (SpotCount Threshold [M03, nucleus, 60]) to visualize the nuclei of cells/events in the G2/M fraction at the single-cell level. A second mask (Area Range [Threshold (M02, tubulin, 75), 50–5000, 0–0.5]) based on the α-tubulin staining (to clearly identify the narrow intracellular bridge of highly condensed α-tubulin that participates to midbody formation) was employed to distinguish telophases from anaphases or cell doublets. Finally, the results from both masks were used to manually verify that selected cells were cells unambiguously in telophase.

### Analysis of cell protein distribution during telophase using Fiji
Staining intensities of α-tubulin, CTV, and of the different markers of interest were analyzed on Fiji to determine the percentage of proteins inherited by the two nascent daughter cells in telophase.

Watershed function of Fiji software was used on the α-tubulin staining intensity to determine the specific areas corresponding to the two nascent daughter cells in telophase. The obtained areas were converted to masks that were next applied to measure CTV and the fluorescence of the different markers of interest. This procedure allowed us to determine the intensity of fluorescence in the two nascent daughter cells in telophase. The percentage of staining in each nascent daughter cell was determined as follows: Fluorescence intensity of daughter cell 1/(fluorescence intensity of daughter cell 1 + fluorescence intensity of daughter cell 2) × 100. To test the specificity of the staining with the different antibodies used to study molecular repartition in telophase, we measured the fluorescent intensity of secondary antibodies or isotype controls as compared to specific antibodies. This analysis gave the following values of mean fluorescence intensity: CD107a 70.527 isotype control 13.621; perforin 716.312, secondary mouse antibody 56.383; GrzB 677.445 isotype control 13.621; T-Bet 356.228 secondary mouse antibody 56.383; and c-Myc 1.434.537 secondary rabbit antibody 14.231.

## 3D laser scanning microscopy on fixed cells
After 72 hr of stimulation, cells were fixed in 1% PFA (10 min, 37°C). Permeabilization and staining with antibodies were performed in PBS 3% BSA, 0.1% saponin (Sigma) for 5 min and 45 min, respectively. The following antibodies were used: anti-human CD107a (10 µg/ml, cloneH4A3, BD Pharmingen) followed by AlexaFluor488 goat anti-mouse IgG1 (10 µg/ml; Invitrogen), anti-human α-tubulin (diluted at 1/100, rabbit polyclonal; Abcam) followed by AlexaFluor555 goat anti-rabbit (10 µg/ml; Invitrogen). Nuclei were labeled with DAPI (1 µg/ml, 5 min). The samples were mounted in 90% glycerol–PBS containing 2.5% DABCO (Sigma) and examined using a LSM710 (Zeiss) confocal microscope with a ×63 plan-Apochromat objective (1.4 oil) with an electronic zoom of 4. Cells in telophase were identified on the basis of nuclear and tubulin marker staining. 3D images (using the

z-stack function) were acquired for every cell identified as being in telophase. CD107a fluorescence intensity in the two nascent daughter cells was measured on 2D image projections obtained applying the Sum function of Fiji Software to z-stack series. Since the background noise made the watershed function unsuitable to use, a region of interest corresponding to the nascent daughter cell was manually drawn on the basis of brightfield and tubulin staining. We determined the percentage of CD107a staining in each nascent daughter cell as follows: CD107a intensity of daughter cell 1/ (CD107a intensity of daughter cell 1 + CD107a intensity of daughter cell 1) $\times$ 100.

## Stimulated emission depletion microscopy

CTL were seeded on poly-L-lysin-coated high-performance coverslips and fixed in 3% PFA (10 min, 37°C). Permeabilization and staining were performed in PBS 3% BSA, 0.1% saponin (Sigma) for 5 min and 60 min, respectively. Cells were stained with an anti-human Granzyme B antibody (10 µg/ml, clone GB11, Thermo Scientific) followed by a goat anti-mouse IgG Abberior Star 580 (Abberior Instruments). Coverslips (high-performance D = 0.17 mm ± 0.005, ZEISS, Germany) were mounted on microscopy slides using Mowiol-DABCO.

STED images were acquired with a Leica SP8 STED 3× microscope (Leica Microsystems, Germany) using a HC PL APO CS2 100×/1.4 oil immersion objective. To optimize resolution without bleaching in 3D, the 775 nm STED laser line was applied at the lowest power that can provide sufficient improvement in resolution compared to confocal. Z-stack series were acquired sequentially with the pulsed 532 nm laser. For image acquisition, we used the following parameters: three time average/line, 400 Hz scan speed. STED images were subsequently deconvoluted with Huygens Professional (SVI, USA) using the CMLE algorithm, with a signal to noise ratio (SNR) of 7. 3D image visualization was performed using the Fiji software.

## Live-cell imaging

For 3D live-cell imaging, the T7 GZMB sequence was obtained by PCR amplification as a XhoI-EcoRI fragment from pCMV-SPORT6-GZMB by using XhoI-T7-GZB forward primer and EcoRI-GRZB noSTOP reverse primer (Employed primers: Name: XhoI-T7-GzB F caaCTCGAGTAATACGACTCAC TATAGGGAGACCCGGTACCatgcaaccaatcctgcttctgcc; Name: EcoRI-GzB-noSTOP-R caaGAATTC cggcgtggcgtttcatggttttctttatccag).

XhoI-EcoRI fragment was cloned as a mCherry-SEpHlurin fusion construct in the pmCherry-SEpHlurin vector to produce the vector pGZMB-mCherry-SEpHluorin available to in vitro *T7 transcription. The plasmid* pCMV-SPORT6-GZMB and pmCherry-SEpHlurin were purchased from Addgene.

For efficient transfection of human CTL with tagged molecules allowing to monitor lytic granule repartition during mitosis, we synthetized capped and tailed poly(A) mCherry-tagged Granzyme B mRNA by in vitro transcription from the plasmid pGZMB-mCherry-SEpHluorin. One microgram of pGZMB-mCherry-SEpHluorin was first linearized by NotI digestion to be used as templates for in vitro transcription by the T7 RNA polymerase using mMESSAGE mMACHINE T7 Ultra kit as per manufacturer's protocol.

Human CTL were transfected using a GenePulser Xcell electroporation system (Bio-Rad). $1 \times 10^6$ CTL (5 days after restimulation therefore in expansion phase) were washed and resuspended in 100 µl Opti-MEM medium (Gibco) at RT with 2 µg mCherry-tagged Granzyme B mRNA (*square wave* electrical pulse at 300V, 2 ms, one pulse). Eighteen hours after transfection, the transfection efficacy was verified by FACS analysis (typically 50–80%). Transfected CTL were seeded into poly-D-lysine-coated eight-well chambered slides (Ibidi, Munich, Germany) before imaging. Chambered slides were mounted on a heated stage within a temperature-controlled chamber maintained at 37°C and constant $CO_2$ concentrations (5%) and inspected by time-lapse laser scanning confocal microscopy (LSM880, Zeiss, Germany, with one image/30 s) for additional 5–6 hr using a Tile Scan mode to enlarge the acquisition fields and capture the rare cells undergoing spontaneous division during the time of acquisition.

For 4D live-cell imaging, 72 hr after stimulation, CD8[+] T cells were stained with Hoechst (200 ng/ml, ThermoFisher Scientific) to sort cells in G2/M phase by flow cytometry (BD FACSAria-SORP, BD Biosciences). Sorted cells were stained with LysoTracker Red (200 nM, ThermoFisher Scientific) for 30 min at 37°C and washed. Twenty thousand cells in 5% HS/IL2/IL15 complete RPMI medium

supplemented with 10 mM HEPES were seeded into poly-D-lysine-coated eight-well chambered slides (Ibidi, Munich, Germany) pre-coated with PDMS micromesh arrays (Microsurfaces, Melburn, Australia) containing 100-µm-diameter wells. Cells were 4D imaged (time and z-stack) on a heated stage within a temperature-controlled chamber maintained at 37°C and constant $CO_2$ concentrations (5%) and inspected overnight by time-lapse laser scanning confocal microscopy with a Plan-Apochromat 40x/1.3 Oil DIC M27 using an LSM780 or LSM880, Zeiss, Germany, or by spinning-disk time-lapse microscopy using a spinning-disk microscope (Nikon) running on Metamorph software. A camera emCCD Evolve (Photometrics) was used for acquisitions. Image analysis was performed using Fiji software, and 4D videos and snapshots were generated with Imaris software.

## Cytotoxicity assay

CTL were incubated with 200 nM LysoTracker Blue a probe staining the acidic lytic compartment of these cells (*Faroudi et al., 2003*) for 30 min at 37°C/5% $CO_2$ in 5% FCS/RPMI/HEPES. After washing, cells expressing the highest and lowest 5–10% LysoTracker Blue staining were sorted using a FAC-SARIA-SORP (BD Biosciences). CTL were used for standard overnight killing assays on the day of cell. Target cells were left unpulsed or pulsed with 10 µM antigenic peptide during 2 hr at 37°C/5% $CO_2$, washed three times, and subsequently transferred to a 96-well U-bottom plate at $10 \times 10^3$ cells/100 µl RPMI, 5% FCS/HEPES. CTL were added to the target cells at the indicated effector (CTL): target (JY) ratio, in 100 µl RPMI, 5% FCS/HEPES. Cells were pelleted for 1 min at 455 g and incubated at 37°C/5% $CO_2$ overnight. Before FACS analysis, 0.25 µg 7-AAD (BD Biosciences) and FITC conjugated anti-CD8 antibody were added to each sample in order to measure the percentage of dead target cells. For the CD107a exposure and CD8 internalization assay, sorted CTL were incubated with target cells at 0.5:1 E/T ratio for 4 hr. Cells were stained with PE-cy7 conjugated anti-CD107a antibody and FITC conjugated anti-CD8 antibody for 30 min in FACS buffer (1% human serum, 1% FCS in PBS), washed, acquired on a Fortessa flow cytometer (BD Biosciences), and analyzed by using FlowJo software (TreeStar).

## Statistical methods

Paired Student's t-test was performed to determine the statistical significance of differences between the groups (GraphPad Prism software version 5).

Chi-square of independence test was performed to determine the independence between the level of expression of a given marker and the capacity of a cell in telophase to asymmetrically distribute this marker (Python software version 3.5).

Kolmogorov–Smirnov goodness-of-fit test was performed to compare law between probability distribution of a marker of interest in cells in G1 (Python software version 3.5).

Chi-square of homogeneity test was performed (in addition Kolmogorov–Smirnov goodness-of-fit test) to determine where the probability distribution of a marker of interest varies (Python software version 3.5).

## Statistical procedures

In the independence chi-square test (*Table 1*), we compare the theoretical effective ($e_{i,j}$) to the observed effective ($n_{i,j}$). The test statistic is defined by:

$$\chi^2 = \sum_{i,j} \frac{(n_{i,j} - e_{i,j})2}{e_{i,j}}$$

We compare it to $\chi^2_{1-\alpha,dl}$, the quantile of the $\chi^2$ distribution associated with the $1-\alpha$ quantile. The quantile with $1-\alpha = 95\%$ is the value such that $P\left(X < \chi^2_{0.95,dl}\right) = 95\%$ where P stands for the probability distribution of the chi-square statistics with the associated degree of freedom dl.

We reject the hypothesis of independence between division of heterogeneous cells and division of all cells in one experiment when $\chi^2 \geq \chi^2_{1-\alpha,dl}$ or when the p-value p satisfies $p < \alpha = 5\%$.

The red boxes represent the situations where we do not reject the hypothesis of independence of division between heterogeneous cells and all cells in one experiment. We shall observe that we never reject the hypothesis of independence.

**Table 1.** Results of independence chi-square test in telophase.

| Independence chi-square test between heterogeneous cells and all cells | Test statistic ($\chi^2$) | $\chi^2_{1-\alpha,\mathbf{dl}}$ | p-value (p) | Degree of freedom (dl) |
|---|---|---|---|---|
| CD107a, Experiment 1, 0 division | 4.060439 | 11.07 | 0.540748 | 5 |
| CD107a, Experiment 1, 1 division | 3.565087 | 11.07 | 0.613563 | 5 |
| CD107a, Experiment 1, 2 divisions | 1.614763 | 7.815 | 0.656047 | 3 |
| CD107a, Experiment 2, 0 division | | | | |
| CD107a, Experiment 2, 1 division | 0.278928 | 7.815 | 0.963942 | 3 |
| CD107a, Experiment 2, 2 divisions | 0.413804 | 7.815 | 0.937376 | 3 |
| CD107a, Experiment 3, 0 division | 2.36867 | 15.51 | 0.967574 | 8 |
| CD107a, Experiment 3, 1 division | 2.092976 | 9.488 | 0.718663 | 4 |
| CD107a, Experiment 3, 2 divisions | 0.655225 | 9.488 | 0.956734 | 4 |

The Kolmogorov–Smirnov test (**Table 2**) is used to define if two independent samples follow the same law, by comparing their cumulative distribution function. We denote the two samples $X_1, X_2, \ldots X_n$ and $Y_1, Y_2, \ldots Y_m$. If we denote by $F_n$ and $F_m$, their cumulative distribution, respectively, the test statistic is defined by:

$$D_{n,m} = x \in R |F_n(x) - F_m(x)|$$

We compare it to $d_{n,m,1-\alpha}$, the quantile of the associated Kolmogorov–Smirnov distribution.

We then reject the hypothesis of adequation between cells of one division and cells of one other division in one experiment when $D_{n,m} \geq d_{n,m,1-\alpha}$ or when the p-value p satisfies $p < \alpha = 5\%$.

The red boxes represent the situation where we do not reject the hypothesis of adequation between cells in one division and cells in another division. The white box represents the situation where we reject this hypothesis.

## Probability of an asymmetric repartition of lytic granules

To obtain a tractable formula for the computation of the computation of the probability of an asymmetric repartition of lytic granules, we use a binomial model. The model postulates that each granule possesses a probability of 0.5 to attain each of the two daughter cells. The binomial model also assumes that all the granules behave independent of each other.

In that case, the probability of an asymmetric division for n granules is then equal to

$$p_n = 2^{-n} \sum_{k < 0.4n} \frac{n!}{k!(n-k)!} + 2^{-n} \sum_{k > 0.6n} \frac{n!}{k!(n-k)!}$$

**Table 2.** Results of Kolmogorov–Smirnov test on G1.

| Experiment 1 | Kolmogorov–Smirnov test | Zero division $D_{n,m}$ | p-value | One division $D_{n,m}$ | p-value | Two divisions $D_{n,m}$ | p-value |
|---|---|---|---|---|---|---|---|
| 1 | 0 division | | | | | | |
| | 1 division | 0.13148 | 0 | | | | |
| | 2 divisions | 0.220034 | 0 | 0.116283 | 0 | | |
| 2 | 0 division | | | | | | |
| | 1 division | 0.087873 | 0 | | | | |
| | 2 divisions | 0.0891924 | 0 | 0.04634 | 0.03582 | | |
| | 3 divisions | 0.054621 | 0.0159 | 0.067702 | 0.001185 | 0.047275 | 0.116534 |
| 3 | 0 division | | | | | | |
| | 1 division | 0.14714 | 0.002607 | | | | |
| | 2 divisions | 0.209553 | 0 | 0.143594 | 0 | | |
| | 3 divisions | 0.190642 | 0 | 0.121757 | 0 | 0.038549 | 0.3545 |

To evaluate the correlation level between particles (between 0 and 1) for a given probability of asymmetric repartition (outside the interval [40–60%]), we use a Monte-Carlo approach where we sampled a sequence of correlated random variables distributed according to a Bernoulli distribution of parameter 0.5 since to the best of our knowledge there is no explicit formula to calculate a such probability of asymmetric repartition. Even with a Monte-Carlo approach, the simulation is a little bit involved: if r is the correlation level and if $X_i$ is the value of the random variable at step i, then $X_{i+1}$ is obtained by:

$$X_{i+1} = X_i Y_i + Z_i(1 - Y_i) \, (Formula \, A)$$

where $Z_i$ is a Bernoulli distribution of parameter 0.5 and $Y_i$ a Bernoulli distribution of parameter r. We shall verify that when $X_1, X_2, \cdots X_n$ are sampled according to Formula A, they are Bernoulli distributed and pairwise correlated with a correlation r. Hence, we then mimic the correlated division with this model and then estimate the probability of asymmetric repartition with 5000 Monte-Carlo simulations for each value of r and a size of n = 90 cells. We then evaluate the desired probability for r varying in a regularly spaced grid from 0 to 1 with a space equal to 0.02.

## Acknowledgements

We thank Dr. Stephane Manenti for discussion and critical reading of the manuscript; Dr. Hellen Robey and Dr. Pauline Gonnord for discussion. We thank Dr. Liza Filali for advice in image analysis and Dr. Juliet Foote for critical reading of the manuscript. We thank the flow cytometry and imaging core facilities of the INSERM UMR 1043, CPTP and of the INSERM UMR 1037, CRCT, Toulouse, France.

This work was supported by grants from the Laboratoire d'Excellence Toulouse Cancer (TOU-CAN) (contract ANR11-LABEX), Region Occitanie (contracts RCLE R14007BB, 671 34 No 12052802, and RBIO R15070BB, No 14054342), Fondation Toulouse Cancer Santé (contract 2014CS044) from the Ligue Nationale contre le Cancer (Equipe labellisée 2018) and from Bristol-Myers Squibb (No CA184-575). RJ was supported by the Ligue Nationale contre le Cancer.

## Additional information

### Funding

| Funder | Grant reference number | Author |
|---|---|---|
| Laboratoire d'Excellence Toulouse Cancer | ANR11-LABEX | Salvatore Valitutti |
| Region occitanie | RCLE R14007BB 671 34 No 12052802 and RBIO R15070BB No 14054342 | Salvatore Valitutti |
| Fondation Toulouse Cancer Santé | 2014CS044 | Salvatore Valitutti |
| Ligue Contre le Cancer | équipe labellisée | Salvatore Valitutti |
| Ligue Contre le Cancer | 4th year phD | Romain Jugele |
| Bristol-Myers Squibb | No CA184-575 | Salvatore Valitutti |

The funders had no role in study design, data collection and interpretation, or the decision to submit the work for publication.

### Author contributions

Fanny Lafouresse, Conceptualization, Formal analysis, Validation, Investigation, Visualization, Methodology, Writing - original draft, Writing - review and editing; Romain Jugele, Conceptualization, Resources, Formal analysis, Investigation, Visualization, Methodology, Writing - original draft, Writing - review and editing; Sabina Müller, Resources, Formal analysis, Investigation, Methodology, Writing - original draft, Writing - review and editing; Marine Doineau, Formal analysis, Writing - review and editing; Valérie Duplan-Eche, Marie-Pierre Puisségur, Resources, Methodology, Writing -

review and editing; Eric Espinosa, Conceptualization, Writing - review and editing; Sébastien Gadat, Conceptualization, Software, Formal analysis, Writing - original draft, Writing - review and editing; Salvatore Valitutti, Conceptualization, Supervision, Funding acquisition, Visualization, Writing - original draft, Writing - review and editing

**Author ORCIDs**
Fanny Lafouresse (iD) https://orcid.org/0000-0001-6572-8631
Salvatore Valitutti (iD) https://orcid.org/0000-0002-6432-4421

**Ethics**
Human subjects: Buffy coats of healthy donors were obtained through the Etablissement Français du Sang (EFS, Toulouse, France). Blood samples were collected and processed following standard ethical procedures (Helsinki 433 protocol), after obtaining written informed consent from each donor and approval by the French Ministry of the Research (transfer agreement AC-2014-2384). Approbation by the ethical department of the French Ministry of the Research for the preparation and conservation of cell lines and clones starting from healthy donor human blood samples has been obtained (authorization No DC-2018-3223).

**Decision letter and Author response**
Decision letter https://doi.org/10.7554/eLife.62691.sa1
Author response https://doi.org/10.7554/eLife.62691.sa2

# Additional files

**Supplementary files**
• Transparent reporting form

**Data availability**
All data generated or analysed during this study are included in the manuscript and supporting files.

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
