## [Decision Letter]

**Acceptance summary:**

Individual cytotoxic T cells differ dramatically in their killing efficiency, but how this heterogeneity is generated is unclear. By analyzing the distribution of the lytic machinery in dividing cytotoxic T cells together with mathematical modeling, Valitutti and colleagues provide evidence that the stochastic asymmetric distribution of cytotoxic granules during cell division makes a substantial contribution to functional heterogeneity amongst the progeny of cytotoxic T cell clones.

**Decision letter after peer review:**

Thank you for submitting your article "Stochastic asymmetric repartition of lytic machinery in dividing CD8^+^ T cells generates heterogeneous killing behavior" for consideration by *eLife*. Your article has been reviewed by three peer reviewers, one of whom is a member of our Board of Reviewing Editors, and the evaluation has been overseen by Tadatsugu Taniguchi as the Senior Editor. The reviewers have opted to remain anonymous.

The reviewers have discussed the reviews with one another and the Reviewing Editor has drafted this decision to help you prepare a revised submission.

Summary:

In this original and interesting study by Valitutti and colleagues, the authors analyzed the distribution of the lytic machinery in dividing cytotoxic T cells. Using a combination of approaches (imaging flow cytometry, confocal imaging and time lapse microscopy), they show that following T cell division, daughter's cells may inherit different numbers of lytic granules. This phenomenon was observed in around 20% of dividing T cells. It is also reported that the granules content correlated with the efficacy of killing suggesting that the asymmetric repartition of the lytic machinery can have functional consequences. While one could argue that the findings are not completely unsurprising (given that the cytotoxic granules are relatively limited in a given T cell), this is a convincing and well performed demonstration that this phenomenon occurs frequently and may contribute to T cell functional diversity. The manuscript is well written and the imaging and time-lapse videos are of high quality. The topic is of considerable interest, both in terms of the significance of understanding CTL functional heterogeneity, and the general question of how biological heterogeneity is generated.

Essential revisions:

1) The author used mathematical modeling to estimate the rate of asymmetric repartition as a function of the number of granules. As expected, the model predicts that the phenomenon is more important with low number of granules. Was that also observed experimentally? In some of the videos, it seems that lytic granules may not behave completely independently from each other. The authors are encouraged to perform quantification and to discuss the relation between the model (that assumes that lytic granules behave independently) and the data. This should be relatively straightforward with the existing data.

2) To provide functional evidence for the physiological significance of the observations, they performed an experiment in which they FACS sorted cells with the highest and lowest content of cytotoxic granules from a CTL clone, and compare their ability to kill target cells in vitro (Figure 7). While these data imply that having more cytotoxic granules may make killing more effective, it was unclear to what extent stochastic segregation mechanisms (as opposed to the degree of CTL activation or differentiation) contribute to the heterogeneity in CTL granule content in this setting. It would also be helpful if the authors could provide a quantitative comparison of the difference in granule content between daughter cells that undergo asymmetric repartition (from the plots in Figures 1, 2, it appears to be in the 1.2 to 2 fold range), and compare this to the difference between the sorted high and low populations from Figure 7 (~2 fold assuming the values provided are based on a linear, not log scale).

---

## [Author Response]

Essential revisions:1) The author used mathematical modeling to estimate the rate of asymmetric repartition as a function of the number of granules. As expected, the model predicts that the phenomenon is more important with low number of granules. Was that also observed experimentally? In some of the videos, it seems that lytic granules may not behave completely independently from each other. The authors are encouraged to perform quantification and to discuss the relation between the model (that assumes that lytic granules behave independently) and the data. This should be relatively straightforward with the existing data.

We agree with reviewers on the point that, in some of the videos, it seems that lytic granules may not behave completely independently from each other. This is probably due to the fact that lytic granules might form transitory aggregates within confined intracellular spaces. We have clarified this point in our revised manuscript (subsection “Lytic granules randomly distribute on the two sides of the cleavage furrow”).

We could not perform precise quantification of this phenomenon with our experimental set up based on confocal and spinning-disk live cell microscopy. To capture rare division events, we need relatively large fields of inspection and long-time periods of acquisition. Therefore, while our videos allow to quantify the integrated fluorescence intensity of mCherry-tagged GrzB or LysoTracker Red in nascent daughter cells, they do not have a high enough resolution and frequency of image acquisition to precisely enumerate individual lytic granules in time and space.

Nevertheless, to further comply with reviewers’ suggestion, we upgraded our mathematical simulation of lytic granule repartition in telophase to include the possibility that lytic granules might not segregate completely independently.

We simulated particle correlation during cell division for 10 to 100 particles, a range that includes our STED microscopy estimation of lytic granule numbers in individual CTL (14-65, mean = 37) and values published in previous studies, now cited in the revised manuscript (Chiang et al., 2017; Clark et al., 2003; Kataoka et al., 1996; Peters et al., 1991). The analysis shows that for a probability of 20% asymmetric repartition of particles (corresponding to 20% uneven repartition of lytic granules during cell division experimentally measured by imaging flow cytometry and confocal imaging), particle correlation has a relatively low value (4% for 37 particles), suggesting that lytic granules mainly segregate independently during cell division. These data are now shown in new Figure 6F.

Taken together, our videos and results in Figure 6 suggest that, in a limited number of division events, granules might not segregate completely independently from each other, implying that some active mechanism of granule segregation might contribute, to a minor extend, to their repartition in telophase. This point is now discussed in the Discussion section.

2) To provide functional evidence for the physiological significance of the observations, they performed an experiment in which they FACS sorted cells with the highest and lowest content of cytotoxic granules from a CTL clone, and compare their ability to kill target cells in vitro (Figure 7). While these data imply that having more cytotoxic granules may make killing more effective, it was unclear to what extent stochastic segregation mechanisms (as opposed to the degree of CTL activation or differentiation) contribute to the heterogeneity in CTL granule content in this setting.

We thank the reviewers for raising this point. Given that lysotracker high and lysotracker low sorted cells are originating from same clonal populations and separated only for the killing assay (over-night), it is unlikely (even if it cannot be excluded) that cells will have different levels of activation/differentiation. Nevertheless, we took up reviewers’ suggestion and performed additional experiments on sorted CTL for high and low LysoTracker staining and measured CD107a surface exposure and CD8 internalization following antigenic stimulation. Results show that LysoTracker^high^ CTL exhibit a higher lytic granule secretion as detected by CD107a exposure when compared to LysoTracker^low^ CTL. However, productive TCR engagement is comparable in both populations as detected by similar levels of CD8 internalization. Together, these results indicate that the lytic granule cargo of individual CTL and not their activation properties directly impact killing behavior. These new results are show in the revised Figure 7 and are described in the subsection “The level of lytic granule content in individual CTL dictates CTL killing capacity”.

It would also be helpful if the authors could provide a quantitative comparison of the difference in granule content between daughter cells that undergo asymmetric repartition (from the plots in Figures 1, 2, it appears to be in the 1.2 to 2 fold range), and compare this to the difference between the sorted high and low populations from Figure 7 (~2 fold assuming the values provided are based on a linear, not log scale).

We agree with the reviewers on the point that it would be helpful to provide a quantitative comparison of the difference in granule content between daughter cells that undergo asymmetric repartition and sorted high and low populations. Nevertheless, we feel that the requested comparison between Figures 1-2 (distribution of CD107a in nascent daughter cells calculated for each cell in telophase at individual cell level by imaging flow cytometry and confocal microscopy) and Figure 7 (FACS analysis of thousands of cells that are either LysoTracker^high^ or LysoTracker^low^ within the whole proliferating T cell population) might be confusing for readers. Indeed, these values are difficult to compare since they have been obtained by using different experimental approaches and different cells.

To comply with reviewer suggestion, we performed this analysis (shown in Author response image 1). Results show that daughter cells that undergo asymmetric repartition and sorted high and low populations exhibit a similar range of differences in granule content. We would prefer to not include this figure in the revised manuscript for the above-mentioned reasons.

**Author response image 1. sa2fig1:** CD107a fold increase measured by comparing the two nascent daughter cells in individual dividing CD8^+^ T cells (Figure 1A and 2A) and Lysotracker fold increase measured by comparing Lysotracker^high^ and Lysotracker^low^ sorted CTL (Figure 7A). Results are from the indicated figures of the revised manuscript. The red line indicates the 1.5 threshold for asymmetric repartition of CD107a (corresponding to the 40-60% range) between daughter cells. Numbers in blue indicate the percentage of individual telophasic cells exhibiting a CD107a fold increase higher than 1.5.